# CLUE-VAD: Structured Semantic Clues for Understanding Explainable Events in Video Anomaly Detection

## Abstract

Weakly Supervised Video Anomaly Detection (WSVAD) aims to identify rare and abnormal events in long untrimmed videos using only video-level labels. While recent approaches have leveraged multimodal learning and pretrained language models, they often treat scenes holistically, failing to provide fine-grained or interpretable insights into the source of anomalies. In this paper, we introduce **CLUE-VAD**, a novel framework that explicitly decomposes each video segment into three semantically grounded components—*Action*, *Environment*, and *Object*—termed as **Textual CLUEs**. This structured decomposition enables the model to disentangle overlapping contextual cues and reason about anomalies in a human-aligned and interpretable manner. Our approach comprises three key modules: (i) the **Witness Module**, which automatically generates dense, clue-specific captions and CLUE-based features using a large-scale video-language model; (ii) the **Detective Module**, which employs a learnable clue-aware fusion mechanism to dynamically quantify the importance of each semantic clue for anomaly prediction; and (iii) the **Reporter Module**, which provides fine-grained explanations by attributing anomaly scores to specific keywords and clues. We also construct the **CLUE-VAD Benchmark**, an enriched evaluation resource with structured segment-level captions for existing WSVAD datasets. Experiments on UCF-Crime and XD-Violence demonstrate that CLUE-VAD achieves strong performance in text-only settings while offering transparent and context-aware anomaly reasoning. Our framework bridges the gap between machine prediction and human interpretation, making it a practical and trustworthy solution for real-world surveillance.

## 1 Introduction

Video Anomaly Detection (VAD) is a core task in intelligent surveillance systems, aiming to automatically identify rare and abnormal events in long, untrimmed video streams. However, collecting temporally localized annotations for anomaly events is expensive and labor-intensive, especially in large-scale real-world scenarios. To alleviate this challenge, Weakly Supervised Video Anomaly Detection (WSVAD) has emerged as a practical and scalable alternative, where only video-level labels, indicating whether a video is normal or abnormal are available during training. This weak supervision setting poses unique challenges in learning fine-grained temporal anomaly patterns, while enabling efficient training on large video corpora.

**What Constitutes a Truly Anomalous Event?** Real-world anomalies are often context-dependent—an event may seem normal in one setting but suspicious in another. Capturing such contextual nuances demands structured reasoning aligned with human perception, requiring detailed understanding of who is involved, what action is occurring, and where the event takes place. For instance, a person running in a park during daytime is typically benign, but the same action with a weapon in a school corridor may signal danger. Such examples illustrate that a fine-grained semantic decomposition is crucial for accurately interpreting anomalies. Motivated by this, we define **Textual CLUEs** as structured semantic components—**Action, Environment, and Object**—that capture the critical dimensions of abnormality. By explicitly decomposing video segments into CLUEs, we aim to improve both the accuracy and interpretability of anomaly recognition. This clue-based repre-

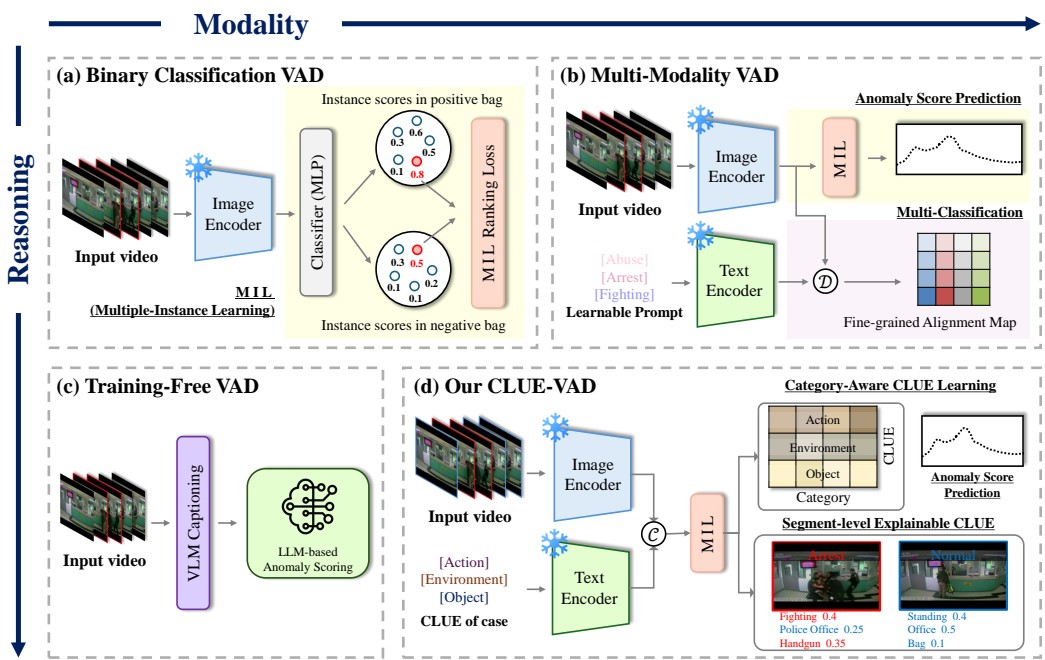

Figure 1: Overview of representative paradigms in WSVAD. (a) **Binary Classification** models use Multiple-Instance Learning to assign anomaly scores based only on visual features, lacking semantic understanding. (b) **Multi-modal VAD** integrates language cues but treats the scene as a whole, limiting interpretability. (c) **Training-Free VAD** generates one caption per segment, often failing to distinguish key semantic factors and limiting fine-grained interpretability. (d) **CLUE-VAD** explicitly decomposes each segment into three semantic clues—*Action*, *Environment*, and *Object*—enabling context-aware and explainable anomaly reasoning.

sentation facilitates a more transparent decision process and aligns closely with how humans assess anomalous situations.

Existing methods often overlook structured decomposition of contextual cues. To clarify recent trends in WSVAD, we categorize prior work into four paradigms (Figure 1). (a) Binary Classification with MIL forms the traditional base: videos are segmented and treated as bags of instances, with anomaly scores predicted via binary classification using only visual features. These methods focus on statistical irregularities in appearance or motion, lacking deeper semantic understanding. (b) Multi-modal VAD enhances visual approaches by incorporating semantic cues from language models, often using vision-language features or prompt tuning. Though effective, these methods treat scenes holistically, limiting fine-grained interpretability. (c) Training-free VAD frames anomaly detection as zero-shot captioning using pretrained video-language models, producing readable summaries without further training. However, by generating one holistic caption per segment, they fail to disentangle semantic elements like agent, action, and location, reducing detailed interpretability.

To bridge these gaps, we introduce (d) **CLUE-VAD**, a novel framework that unifies fine-grained anomaly detection, interpretable reasoning, and benchmark-level evaluation. We first construct a the **CLUE VAD Benchmark** that enriches existing weak supervision datasets with structured, segment level captions. Additionally, we provide a strong **Baseline model** that decomposes each video segment into three human readable clues, using a pretrained video language model. These clues align with how people judge abnormality and are fused in a category-aware scoring pipeline that both boosts accuracy and highlights which clue drives each anomaly. Keyword-level attribution within every clue offers transparent, segment-wise explanations, converting raw scores into actionable insights. Taken together, the benchmark, baseline, and reasoning pipeline make CLUE-VAD a practical and trustworthy solution for real-world surveillance.

This clue-based decomposition boosts both detection accuracy and interpretability. By encoding semantic clues separately, the model attributes abnormal evidence to the most influential role and surfaces keywords: **Environment** terms (e.g., "fire," "smoke") raise scores in Explosion, **Object** words

("knife," "gun") dominate Robbery, and **Action** words ("punch," "kick") are decisive in Fighting. Such fine-grained attribution enables reliable, transparent reasoning for real-world surveillance.

**Our contributions are summarized as follows:**

- We propose **CLUE-VAD**, the first video anomaly detection framework that explicitly decomposes each video segment into structured, clue-specific textual descriptions—*Action*, *Environment*, and *Object*—to enable semantically grounded and context-aware reasoning.

- We develop the **Witness Module**, a scalable and automated captioning pipeline that generates dense descriptions for both normal and abnormal segments. It further extracts CLUE-based features that form the foundation for semantic-level anomaly analysis.

- We introduce the **Detective Module**, which incorporates a learnable clue-aware fusion mechanism to dynamically estimate the relative importance of each semantic clue for anomaly prediction. This enables interpretable decision-making via score decomposition and attention-based visualization.

- We show that CLUE-VAD achieves competitive performance in text-only settings on two large-scale benchmarks—UCF-Crime and XD-Violence. Furthermore, the **Reporter Module** offers fine-grained, human-aligned explanations by attributing anomaly scores to specific keywords and semantic clues, demonstrating the practical utility of our framework for **explainable video anomaly detection**.

## 2 RELATED WORK

**Weakly supervised VAD.** Early WSVAD Feng et al. (2021) Cho et al. (2023) Cho et al. (2024) follows the MIL Sultani et al. (2018) paradigm: treating a video as a bag of temporal snippets, with variants such as RTFM Tian et al. (2021) and WSAL Lv et al. (2021) that model temporal consistency for better localization. Recent work LEC-VAD Wang & Chen (2025) focuses explicitly on event completeness, introducing a dual (category agnostic/aware) branch and a Gaussian-mixture prior to smooth scores and recover contiguous anomalous intervals, thereby addressing fragmented predictions common in MIL pipelines. Our CLUE-VAD is orthogonal: rather than changing the scoring dynamics, we restructure the clue used by the scorer, decomposing each segment into Action, Environment, and Object clues and learning their category-aware importance under weak supervision. This yields competitive detection while enabling attribution without extra labels.

**Vision–language models for VAD.** VLM-based methods inject high-level semantic textual information into WSVAD Chen et al. (2023) Zanella et al. (2024) Sun et al. (2024) Appiani & Beyan (2024) Li et al. (2025). For instance, VadCLIP Wu et al. (2024) adapts CLIP Radford et al. (2021) with learnable textual prompts built around a hand-picked anomaly keyword which is fixed but the sentence context is learned, rather than relying on fully hand-crafted prompts. Prompt engineering variants (PEMIL Chen et al. (2024), TPWNG Yang et al. (2024)) and memory-based text enrichment (LECVAD Wang & Chen (2025)) further refine the alignment of the text-vision. Our approach differs in structure: instead of a single holistic prompt per segment, proposed CLUE-VAD builds three clue-aligned textual streams and performs clue-aware fusion, producing per-clue scores and keywords that directly explain predictions.

**Explainable VAD and open-world understanding.** In VAD, explainability has been pursued from multiple angles. EVAL Singh et al. (2023) explains anomalies via an "instrument panel" showing motion direction (angle), speed, and object presence. But its motion-centric design struggles with non-motion-dominant anomalies such as shoplifting, limiting generality. HSC Sun & Gong (2023) emphasizes scene-awareness by contrasting scene- and object-level features learned only from normal data. In open-world VAD, HAWK Tang et al. (2024) faces shifting anomaly taxonomies and severe long-tail sparsity, while relying on VLM priors that are prompt-sensitive and prone to bias/hallucination. These factors hinder robust temporal grounding and score calibration across categories, limiting reliability. Holmes-VAU Zhang et al. (2025) introduces HIVAU-70k an anomaly-focused temporal sampler for long-term reasoning. However, these approaches cannot perform category-aware, role-disentangled reasoning or learn per-category clue fusion, limiting fine-grained interpretability. Sherlock Ma et al. (2025) relies on spatially factorized experts stacked on many pretrained modules and a two-stage mix of external datasets. These make the pipeline heavy and data-dependent potentially hindering efficiency and portability. Our novelty is to bring

a three-clue decomposition into WSVAD, learn per-category clue weights with MIL, and report per-clue attributions. This yields transparent, clue-specific reasoning and complements temporal regularization methods.

## 3 CLUE-VAD BENCHMARK

To enable interpretable and clue-aware video anomaly detection, we construct a new textual description built upon existing video anomaly datasets. Specifically, we utilize the UCF-Crime dataset, which is a large-scale untrimmed benchmark commonly used in weakly supervised video anomaly detection. Our description pipeline transforms unstructured video content into structured and semantically aligned textual clues, which we refer to as CLUEs. This pipeline is designed to be generalizable and can be applied to other video anomaly detection datasets, enabling scalable and flexible extraction of textual descriptions. The CLUE contains three semantic information: action, environment, and object. Please refer to the Appendix for a detailed description about benchmark.

**Pipeline of CLUE Captioning.** To generate structured textual representations aligned with human reasoning, we construct a captioning pipeline that decomposes each video segment into three semantically distinct CLUEs. This process utilizes InternVideo 2.5, a large-scale model generating context-aware video descriptions. Given an untrimmed surveillance video, we first divide it into non-overlapping video segments of 64 frames. Each segment serves as the input unit for captioning. For every segment, we issue three separate clue-specific prompts to captioning model in order to extract complementary information. The prompts are designed to isolate the functional components of an event. The action prompt requests a description of the main activity taking place in the scene. The environment prompt elicits information about the spatial or contextual setting of the event. The object prompt asks the model to enumerate entities that are involved or interacted with. These prompts guide captioning model to attend to distinct semantic aspects of each scene, enabling the construction of multi-perspective captions that play a pivotal clue in detecting abnormal events by offering diverse contextual cues essential for nuanced interpretation of real-world video content. The generated captions are organized such that each entry corresponds to an individual video segment and encapsulates all three clue-driven descriptions.

**Semantic Analysis of CLUE-Based Descriptions.** To quantify the information carried by each CLUE, we extract the top–$k$ keywords from every action, environment, and object caption using LLaMA3 Dubey et al. (2024). The keywords are then pooled by anomaly category and by label (normal *vs.* abnormal). Complete frequency tables are provided in Appendix, while below we discuss the most salient shifts. In action, abnormal segments contain violent verbs such as *"restraining"*, *"escorting"*, *"attacking"* and *"fighting"*. For Environment, Explosion and Arson, abnormal clips refer to environments like *"a burning public bus"* or *"a warehouse on fire"*. Likewise, object captions in Burglary and Shooting emphasise weapons (e.g., *"a black rifle"*, *"a black handgun"*), while Shoplifting clips highlight innocuous items like *"a tablet"*. These systematic, clue-specific shifts confirm that the CLUE templates capture discriminative semantics that are strongly aligned with anomaly context.

**Toward Interpretable clue-Aware descriptions.** The structure of CLUE-VAD captions enables new forms of semantic supervision that are not possible with traditional flat captions. Unlike previous approaches that generate a single textual description per video or segment, CLUE-VAD provides explicit and disentangled clue descriptions across three semantic clues. This allows the models to learn from each component independently. This separation not only enhances the performance of anomaly detection, but also facilitates interpretable diagnostics, clue-specific ablation studies, and explanation generation. In addition, the structured CLUE format is readily compatible with a wide range of downstream video-language tasks. For example, in text-based video retrieval, each clue-specific clue can serve as a targeted query (e.g., *"person running in a subway station"*) to retrieve temporally relevant segments, even in long untrimmed videos. In zero-shot anomaly detection or classification, the semantic clues can be composed into prompt-like descriptions that generalize to unseen event types without additional training, by leveraging pretrained vision-language models. Furthermore, the clue-aligned structure supports sentence grounding, event forecasting, and cross-modal reasoning, making CLUE a versatile and extensible resource beyond its original anomaly detection objective.

Figure 2: CLUE-VAD baseline framework. It consists of three modules: (a) **Witness** extracts clue-specific features using a video-language model, (b) **Detective** performs category-aware anomaly prediction with MIL loss, and (c) **Reporter** provides interpretable scores via clue-level attention for segment-wise attribution.

# 4 CLUE-VAD BASELINES

Human visual perception considers multiple contextual cues when identifying real-world complex anomaly scenarios. For example, a scene may appear anomalous not solely due to the action itself, but also because of the object involved or the surrounding environment. To reflect this human-aligned understanding, we decompose each event into three semantic dimensions: Action (Act), Environment (Env), and Object (Obj). Instead of relying on holistic video-level captions, we extract clue-specific descriptions and leverage clue-aware feature learning to enhance interpretability and support fine-grained anomaly reasoning.

Baseline of CLUE-VAD is a novel interpretable video anomaly detection framework that incorporates structured textual cues to improve both detection performance and explainability. Figure 2 illustrates the overall pipeline, which consists of three core modules. **(a) Witness Module** extracts clue-wise features using clue-specific captions through a pretrained video-language model, enabling semantic decomposition of each event. **(b) Detective Module** then performs anomaly prediction using category-aware clue learning, where learnable weights determine the contribution of each clue for different anomaly types. This branch is optimized for prediction accuracy under weak supervision using MIL loss. For interpretation ability, **(c) Reporter Module** applies segment-wise clue attention to highlight semantically meaningful clues. With clue-attention factor, it generates explainable anomaly scores and identify temporally salient segments. Together, these modules bridge low-level video understanding and high-level anomaly reasoning, enabling both strong detection performance and post-hoc interpretability.

## 4.1 WITNESS MODULE

Given an input video, we uniformly divide it into $T$ non-overlapping temporal segments, where each segment covers 16 frames. For every segment, we generate three separate textual captions, each targeting a different semantic clue. Once these captions are obtained, we encode them into CLUE-wise feature vectors using a frozen CLIP text encoder $f_{\text{text}}(\cdot)$. These textual embeddings serve as the clue-specific representations of each segment, which will later be used for both prediction and explanation. In parallel, we extract a visual embedding $\mathbf{v}^{(t)}$ for each segment using a frozen CLIP visual encoder $f_{\text{visual}}(\cdot)$.

$$\mathbf{c}_r^{(t)} = f_{\text{text}}(D_r^{(t)}), \quad \mathbf{v}^{(t)} = f_{\text{visual}}(I^{(t)}), \quad \text{where } t \in \{1, 2, ..., T\} \tag{1}$$

Here, $\mathbf{c}_r^{(t)} \in \mathbb{R}^d$ denotes the text embedding corresponding to clue $r \in \{act, env, obj\}$ for $t$-th segment. $D_r^{(t)}$ is the text description generated for that CLUE, and $I^{(t)}$ is the image (keyframe) corresponding to each segment. This process ensures that each segment is represented by three distinct, clue-aware embeddings that capture complementary semantic context.

## 4.2 DETECTIVE MODULE

The contribution of each semantic clue to anomaly recognition varies across event categories. For instance, in the category "Arrest", action-related clue such as "tackle" or "struggle" tend to be more indicative than environmental context. In contrast, for "Explosion" events, environmental elements like "fire" or "smoke" often play a more critical clue than the associated actions or objects. To accommodate such category-specific variability, we design our model to learn the relative importance of each clue in a category-aware manner. This category-aware clue learning choice enables more adaptive and context-sensitive anomaly detection.

Concretely, for each video category $c \in \mathcal{C}$, we define a category-aware Learnable weights $w_{Act}^{(c)}, w_{Env}^{(c)}, w_{Obj}^{(c)}$, where each element corresponds to the importance score of Action, Environment, and Object. These weights are passed through a softmax function to ensure that they form a valid distribution, enabling stable training and interpretability.

To incorporate the relative importance of each clue for a given category $c$, we apply Category-aware CLUE Learning to $w_r^{(c)}$. These weights are used to modulate the clue embeddings and their concatenation through a linear transformation:

$$\mathbf{E}_r^{(c)} = \mathbf{W}_{\text{proj}}^\top \cdot \left( \left( w_r^{(c)} \odot \mathbf{x}_r^{(t)} \right) \| \mathbf{v}^{(t)} \right) \tag{2}$$

where $\odot$ denotes denotes the clue-wise scaling and concatenation, $\|$ indicates vector concatenation. $\mathbf{W}_{\text{proj}} \in \mathbb{R}^{d \times 3d}$ is a projection matrix applied through an MLP. The final CLUE-wise feature is then obtained by concatenating with the visual embedding $\mathbf{v}^{(t)}$, forming a unified multimodal representation for each segment. This fused feature $\mathbf{E}_r^{(c)}$ captures the segment content in a category-sensitive manner, emphasizing the most relevant semantic clues for each category. As the model is trained under the MIL framework, the resulting clue weights $w_r^{(c)}$ modulate the relative importance of each semantic clue type. These weights are continuously refined through backpropagation and serve as the foundation for our **Category-Aware CLUE Learning** mechanism, where the model learns to emphasize clues that are most relevant to each category.

To support this process, we maintain a Case Log Embedding that stores updated clue weights per category, indexed by video category $c$ and clue type $r$ for precise retrieval and interpretability. To obtain an accurate category for selecting CLUE weights, the Witness module queries InternVideo to extract category cues (e.g., caption-derived keywords), which are injected into the concatenated clue features before classification. During training, the model uses the ground-truth category to fetch the corresponding weights from the log; at inference, it uses the predicted category to retrieve them. The predicted category is produced by a category classifier trained with cross-entropy loss on the concatenated clue features.

Finally to obtain the anomaly score $\hat{s}_t$, we pass this feature through an MLP-based scoring function $\phi(\cdot)$, composed of two fully connected layers with ReLU and Sigmoid activations. This mechanism allows the model to dynamically select category-aware clue weights at test time and ensures that the learned fusion strategy remains consistent across training and inference. Moreover, the case log supports interpretability and enables post-hoc analysis by preserving the learned clue-specific importance for each category.

## 4.3 REPORTER MODULE

While Category-Aware CLUE Learning enables accurate anomaly prediction, it does not offer insights into why a particular segment is considered anomalous. To achieve interpretability at the segment level, we introduce a attention mechanism that dynamically weights the importance of each semantic clue for every temporal segment. This allows the model to adapt its focus based on local variations in clue saliency and generate explanation-aware scores.

Given a $t$-th segment, we compute the attention score $a_r^{(t)}$ for each clue $r \in \{Act, Env, Obj\}$ by applying a learnable linear projection followed by a ReLU activation. The output is then projected by a clue-specific query vector $\mathbf{q}_r \in \mathbb{R}^d$, as follows:

$$\alpha_r^{(t)} = \mathbf{q}_r^\top \cdot \text{ReLU}(a_r \mathbf{x}_r^{(t)} + \mathbf{b}_r) \tag{3}$$

Here, $\mathbf{x}_r^{(t)} \in \mathbb{R}^d$ is the clue-specific feature of $t$-th segment, and $a_r \in \mathbb{R}^{d \times d}$ and $b_r \in \mathbb{R}^d$ are learnable parameters for each clue. We then normalize the attention scores across clues using the softmax function. These attention weights $\alpha_r^{(t)}$ indicate the relative importance of each clue at $t$-th segment. The final fused feature for this segment is computed as:

$$z_t^{(\text{attn})} = \sum_{r \in \{Act, Env, Obj\}} \alpha_r^{(t)} \cdot \mathbf{x}_r^{(t)} \tag{4}$$

The fused feature $z_t^{(\text{attn})}$ is then passed into the anomaly scoring network $\phi(\cdot)$, yielding a raw anomaly activation score $\hat{s}_t$. To enhance the visibility of temporal outliers, we apply z-score normalization over the scores of all $T$ segments in a video. Specifically, we compute the mean $\mu$ and the standard deviation $\sigma$. The final normalized anomaly score is then calculated as $s_t^{(z)} = \frac{\hat{s}_t - \mu}{\sigma}$, which reflects the degree of abnormality for $t$-th segment relative to the overall context of the video.

This score $s_t^{(z)}$ reflects the degree of abnormality for $t$-th segment relative to the overall video context. Unlike the scalar-based anomaly score used for model training, this Segment-level Explainable CLUE is employed for post-hoc explanation reasoning and visualization. It facilitates the identification of the most influential clues and enables transparent reasoning for real-world applications.

## 4.4 Loss Function and Regularization

**MIL Ranking Loss.** To supervise the Category-Aware CLUE Learning and Segment-level Explainable CLUE for weakly supervised anomaly detection, we adopt a top-$k$ Multiple Instance Learning (MIL) ranking loss. This formulation assumes that each video is a bag of segments, where a positive (abnormal) video should contain at least $k$ highly abnormal segments, and a negative (normal) video should contain only low-scoring segments.

Let $P$ and $N$ denote the sets of segments in anomalous and normal videos, respectively. Using the predicted anomaly score $\hat{s}_t$ computed from the category-aware Scalar Fusion branch, we compute the average of the top-$k$ segment scores for each bag. The top-$k$ MIL loss is then defined as:

$$\mathcal{L}_{\text{MIL}} = \max\left(0, \ 1 - \frac{1}{k}\sum_{t \in \text{Top-}k(P)} \hat{s}_t \ + \frac{1}{k}\sum_{t \in \text{Top-}k(N)} \hat{s}_t\right) \tag{5}$$

This margin-based objective encourages the model to assign higher anomaly scores to the top-$k$ segments in abnormal videos than to the top-$k$ segments in normal ones. Notably, only the scalar fusion scores are used during training, while the attention-based z-score branch is employed solely for post-hoc explanation and visualization.

**Regularization Terms.** To improve training stability and enhance interpretability of the attention-based branch, we apply two regularization losses on the segment-wise clue attention weights $\alpha_r^{(t)}$. To avoid overconfidence in clue selection and encourage diversity, we minimize the negative entropy of the attention distribution called Attention Entropy Loss $\mathcal{L}_{\text{AE}}$:

$$\mathcal{L}_{\text{AE}} = \frac{1}{T}\sum_{t=1}^{T} \sum_{r \in \{Act, Env, Obj\}} \alpha_r^{(t)} \log \alpha_r^{(t)} \tag{6}$$

Also, to enforce consistency across neighboring segments and reduce abrupt changes in attention, we apply a Temporal Smoothness Loss $\mathcal{L}_{\text{smooth}}$:

$$\mathcal{L}_{\text{smooth}} = \frac{1}{T-1}\sum_{t=1}^{T-1} \sum_{r \in \{Act, Env, Obj\}} \left\| \alpha_r^{(t)} - \alpha_r^{(t+1)} \right\|^2 \tag{7}$$

The complete training objective is $\mathcal{L}_{\text{total}} = \mathcal{L}_{\text{MIL}} + \lambda_{\text{AE}}\mathcal{L}_{\text{AE}} + \lambda_{\text{smooth}}\mathcal{L}_{\text{smooth}}$ where $\lambda_{\text{AE}}$ and $\lambda_{\text{smooth}}$ are regularization coefficients. These regularizers encourage the model to learn smoother and more interpretable attention distributions while maintaining robust anomaly prediction performance.

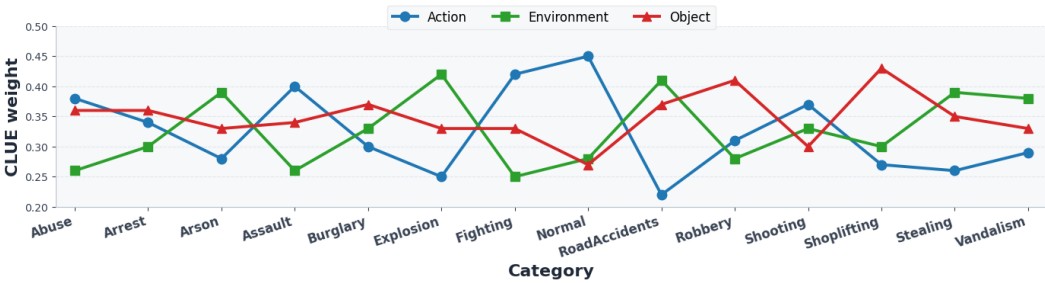

Figure 3: Category-wise distribution of clue importance, showing the relative contribution of *Action*, *Environment*, and *Object* across different anomaly types.

## 5 EXPERIMENT

**Impacts of Category-aware CLUEs.** Figure 3 presents the category-specific CLUE weights learned by the category-aware fusion module on UCF-Crime. *Fighting*, *Assault*, and *Abuse* assign higher weights to Action clues, reflecting their motion-centric nature. In contrast, object-centric events—*Robbery*, *Shoplifting*, *Stealing*—emphasize Object clues, while *Explosion* and *RoadAccidents* rely more on Environment context. Notably, *Normal* also shows strong Action weighting, likely due to subtle routine movements. These patterns confirm that CLUE-VAD adaptively focuses on the most relevant semantic clues per category, enhancing performance and interpretability, and improving robustness across diverse anomaly types.

**Comparisons with VAD methods.** Table 1 compares CLUE-VAD with state-of-the-art methods on UCF-Crime (AUC) and XD-Violence (AP). Our CLUE-based model scores 87.76 / 86.47, surpassing CLIP-based baselines TSA (87.58 / 82.17) and UMIL (86.75 / –), showing the benefit of clue-disentangled supervision and category-aware fusion. Among explainable methods, the full CLUE-VAD model reaches 89.23 / 87.24, outperforming Ex-VAD (88.29 / 86.52) on both datasets and Holmes-VOU (88.96 / 87.68) on UCF-Crime, while remaining within 0.5 AP on XD-Violence. These gains come from (i) category-conditioned fusion over Action/Environment/Object clues, (ii) clue-level disentangled supervision that tightens text–video alignment, and (iii) streaming Top-K, which stabilizes category estimates and focuses learning on informative frames. Overall, CLUE-VAD delivers higher accuracy than prior explainable models without sacrificing transparency, supporting real-world, interpretable VAD.

| Model | Feature | UCFC AUC | XD AP |
|---|---|---|---|
| *SoTA with single-modality* | | | |
| MIL Sultani et al. (2018) | C3D | 75.41 | 75.68 |
| RTFM Tian et al. (2021) | I3D | 84.30 | 77.81 |
| CLAV Cho et al. (2023) | I3D | 86.10 | - |
| UMIL Lv et al. (2023) | CLIP | 86.75 | - |
| UR-DMU Zhou et al. (2023) | I3D | 86.97 | 81.77 |
| TSA Joo et al. (2023) | CLIP | **87.58** | **82.17** |
| *SoTA with Multi-modality* | | | |
| PEMIL Chen et al. (2024) | I3D+Text | 86.83 | 88.21 |
| TPWNG Yang et al. (2024) | CLIP | 87.79 | 83.68 |
| VadCLIP Wu et al. (2024) | CLIP | 88.02 | 84.15 |
| LEC-VAD Wang & Chen (2025) | I3D+CLIP | 89.97 | **88.47** |
| $\pi$-VAD Majhi et al. (2025) | I3D(test) | **90.33** | 85.37 |
| *SoTA with Explainable VAD* | | | |
| LAVAD Zanella et al. (2024) | ViT | 80.28 | 62.01 |
| VERA Ye et al. (2025) | - | 86.6 | **88.2** |
| VADOr Lv & Sun (2024) | - | 88.1 | - |
| Ex-VAD Huang et al. | CLIP | 88.29 | 86.52 |
| Holmes-VAU Zhang et al. (2025) | ViT | 88.96 | 87.68 |
| **CLUE-VAD(Ours)** | ViT-14/H | **89.23** | 87.64 |

Table 1: Comparison with state-of-the-art methods on UCF-Crime (AUC %) and XD-Violence (AP %).

**Anomaly score graph.** To qualitatively assess behavior, Figure 4 visualizes frame-level anomaly scores for five UCF-Crime category. Each column is one video with normal and anomalous segments: the top row shows sampled frames, the bottom row the predicted scores over time. Red bars indicate predicted anomaly intensity, and blue dashed lines mark ground-truth intervals. Scores rise sharply during anomalous events and remain low otherwise; even for ambiguous cases like *Stealing*, the highlighted window closely matches the ground truth. These examples demonstrate temporally precise, interpretable predictions with few

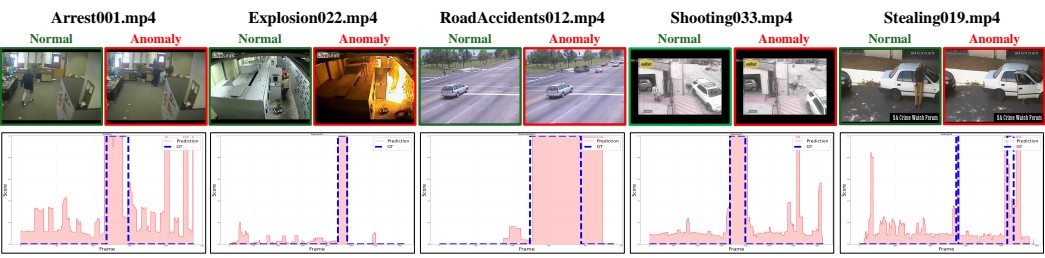

Figure 4: Frame-level anomaly prediction results over time.

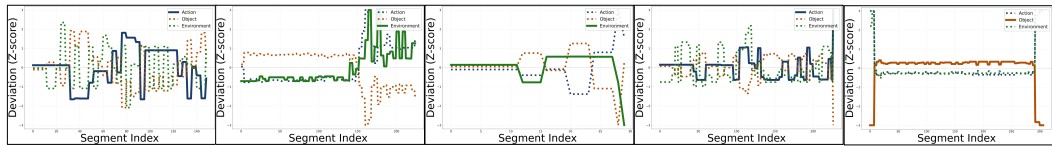

Figure 5: Segment-level interpretability achieved through CLUE-wise score decomposition, highlighting which semantic clue contributes most to each anomaly.

false positives, supporting real-world applicability. For more qualitative plots, please see Appendix Figure S4.

**Segment-level Explainable CLUE score.** Figure 5 plots segment-level z-score deviations for the three clues—*Action*, *Object*, *Environment*—across five categories. In *Explosion022* and *RoadAccidents012*, *Environment* spikes near anomaly boundaries, reflecting abrupt scene changes. For *Arrest001* and *Shooting033*, sharp *Action*/*Object* deviations correspond to rapid human motion and interactions. In *Stealing019*, a sustained rise in *Object* deviation marks the prolonged presence of suspicious items. These role-specific patterns provide interpretable anomaly evidence and align with ground-truth windows, indicating temporally precise role-wise attention—clarifying not only when anomalies occur but also which semantic factors drive them.

**Effectiveness of CLUE Features.** Table 2 shows that each clue adds distinct, complementary signal. Used alone, they emphasize human motion, contextual background, or interacting objects; combined, performance improves, with AUC rising from single to dual and then triple configurations. This validates decomposing video semantics into structured roles and dynamically fusing them, enabling CLUE-VAD to deliver more accurate and interpretable predictions in real deployments at scale.

| Combinations | Action | CLUEs Environment | Object | AUC(%) |
|---|---|---|---|---|
| Single | ✓ | | | 85.72 |
| | | ✓ | | 85.95 |
| | | | ✓ | 85.37 |
| Dual | ✓ | ✓ | | 86.46 |
| | | ✓ | ✓ | 86.50 |
| | ✓ | | ✓ | 86.84 |
| Triple | ✓ | ✓ | ✓ | 87.76 |

Table 2: Effectiveness of each CLUE feature combinations.

**Modality Analysis.** Table 3 shows that text-only and visual-only settings achieve comparable performance, while combining both modalities yields clear gains on UCF-Crime and XD-Violence. Textual CLUEs provide high-level semantic structure (who, what, where), whereas visual features contribute fine-grained appearance and motion cues, making their roles complementary rather than redundant. Overall, these results support our design choice of a multimodal CLUE-VAD, where the synergy between text and vision consistently translates into improved anomaly detection performance.

| Modality | UCF-Crime AUC(%) | XD-Violence AP(%) |
|---|---|---|
| Text only (CLUE) | 87.76 | 86.47 |
| Visual only | 87.46 | 86.35 |
| Multimodal | 89.23 | 87.64 |

Table 3: Performance of CLUE-VAD under different modality.

**Detective Module Interaction Mechanisms.** Table 4 compares several CLUE interaction strategies within the Detective module under the same text-only UCF-Crime setting. Replacing our weighting MLP with a single-layer self-attention mech-

| CLUE Interaction Method | UCF-Crime (AUC(%)) |
|---|---|
| Attention-based fusion | 86.89 |
| Graph-based fusion | 86.13 |
| Pairwise inter-CLUE fusion | 87.46 |
| Category-aware CLUE weighting (ours) | **87.76** |

Table 4: CLUE interaction mechanisms in the Detective module.

anism yielded only 86.89 AUC, indicating that simply treating the three CLUEs as tokens does not effectively capture their distinct roles. A graph-based variant with one-step message passing improved performance to 86.13 AUC but added parameters without meaningful gain. Pairwise CLUE interactions (Act–Env, Env–Obj, Act–Obj) performed better, reaching up to 87.46 AUC, suggesting that certain CLUE pairs provide useful relational cues. Still, the proposed category-aware CLUE weighting achieved the best result at 87.76 AUC by directly modeling category-specific importance across all three CLUEs. Overall, these results show that while alternative interaction designs are feasible, category-aware weighting offers the most robust and parameter-efficient fusion strategy.

**Domain Robustness of the Witness Module.** Table 5 reports cross-dataset evaluation results that demonstrate the robustness of the Witness Module and the overall CLUE-VAD framework under out-of-distribution (OOD) domain shifts. Because the Witness Module relies on structured CLUE extraction rather than holistic cap-

| Source | Target (AUC(%)) | | | |
|---|---|---|---|---|
| | XD | DoTA | CCTV-Fights | UBI-Fights |
| UCFC | 85.73 | 84.59 | 79.23 | 82.34 |

Table 5: Cross-dataset transfer performance from UCF-Crime to different target datasets.

tions, it maintains stable performance across diverse environments, including indoor surveillance scenes, outdoor streets, and dash-cam–style traffic videos. We further validate generalization on multiple heterogeneous datasets, XD-Violence, DoTA Yao et al. (2020), CCTV-Fights Perez et al. (2019), and UBI-Fights. This observe that CLUE-VAD consistently preserves its Action–Environment–Object decomposition even as the visual distribution changes. In addition, the CLUE formulation naturally supports domain-adaptive extensions (e.g., redefining CLUEs as "vehicle context," "road agent," or "driver behavior" for traffic scenarios), highlighting CLUE-VAD as a flexible and extensible paradigm for explainable video anomaly detection in a wide range of real-world domains.

**Efficiency of the CLUE-VAD.** Table 6 summarizes the computational cost of each module. The only expensive operation is caption prompting with a large video–language model (59.57,ms, 20.38,GB), which is executed offline and therefore does not affect real-time deployment. In con-

| Step | Time [ms] | Mem [GB] |
|---|---|---|
| Caption prompting | 59.57 | 20.38 |
| CLUE encoding | 0.64 | 4.35 |
| Category-Aware CLUE Learning | 0.01 | 0.42 |
| Segment-level Explainable CLUE | 0.01 | 0.53 |

Table 6: Inference efficiency of CLUE-VAD.

trast, all online components—CLUE encoding, Category-Aware CLUE Learning, and Segment-level Explainable CLUE—are extremely lightweight, requiring only 0.01–0.64,ms and less than 1,GB of memory. This design ensures that CLUE-VAD delivers CLUE-wise interpretability while maintaining sub-millisecond inference latency suitable for real-time video anomaly detection.

## 6 CONCLUSION

We introduced **CLUE-VAD**, a novel and interpretable framework for WSVAD. By decomposing each video segment into structured semantic clues—*Action*, *Environment*, and *Object*—our method enables fine-grained, context-aware reasoning that aligns with human perception. We also construct the **CLUE-VAD Benchmark**, which enriches existing datasets with segment-level, clue-specific captions to support both training and evaluation. Through its modular design—Witness, Detective, and Report—CLUE-VAD not only achieves strong performance in text-only settings but also provides transparent, clue-level explanations. Our work offers a new paradigm for explainable anomaly detection in real-world surveillance scenarios, while remaining robust and scalable.

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

## A  APPENDIX

## 1. OVERVIEW

This Appendix is organized into three sections that expand on the main paper's methodology, re-sources, and additional results. **Section B (Experimental Setup)** details every element required to reproduce our experiments. We first clarify the evaluation metrics and report the hardware/software configuration together with all hyper-parameter values. Next, we describe how InternVideo 2.5 is used to generate clue-specific captions and how CLIP encoders extract the text and visual features that feed our model. **Section C (CLUE-VAD Benchmark)** documents the construction of the new benchmark. We provide the exact InternVideo 2.5 prompt templates (Figure S1), sample caption outputs (Figure S3), the Llama 3 keyword-extraction prompt (Figure S2), and the resulting key-word statistics (Table S1, S2, S3). The section concludes with the JSON file format that stores per-segment keywords and deviation scores, enabling external use of the benchmark. **Section D (CLUE-VAD Baseline)** complements the main paper by visualizing the model's behaviour and re-porting extended metrics. Specifically, we plot frame-level anomaly scores for additional videos (Figure S4) and list category-wise AUC/AP values on UCF-Crime and XD-Violence (Table S4), illustrating the robustness of our approach across event types.

## 2. EXPERIMENTAL SETUP

**Benchmark Datasets and Anomaly Definitions.**  We evaluate CLUE-VAD on two large-scale public benchmarks widely adopted in Video Anomaly Detection (VAD). UCF-Crime (UCFC) Sul-tani et al. (2018) consists of 1,900 long, untrimmed surveillance videos captured from real-world CCTV cameras, covering 13 diverse categories of abnormal events including *Robbery*, *Assault*, and *Explosion*. All videos are annotated with video-level labels indicating whether an anomaly occurs, while temporal annotations of anomalies are provided only for the test set. Due to its scale and va-riety, UCFC has become a standard benchmark for weakly-supervised VAD methods. XD-Violence (XD) Wu et al. (2020) is a large-scale anomaly dataset comprising 4,754 untrimmed videos collected from diverse sources such as surveillance, dash-cams, and mobile cameras. It spans 6 anomaly classes with varying forms of violent behaviors including *Fighting*, *Road Accident*, and *Explo-sion*. XD-Violence provides frame-level annotations for both training and testing sets, enabling a more granular assessment of anomaly localization performance compared to UCFC. Together, these datasets allow comprehensive evaluation of CLUE-VAD under diverse real-world anomaly detection scenarios, demonstrating its effectiveness in generalizing across different anomaly types, camera se-tups, and environments.

**Hyperparameter Tuning.**  All experiments are evaluated with frame-level AUC and AP. Train-ing and inference are carried out on a single NVIDIA RTX3090 GPU (24GB) using automatic mixed precision. Each video is split into 16-frame segments, and every mini-batch contains an equal number of normal and abnormal segments. The text-only configuration is trained for 40 epochs, whereas the multi-modal variant is trained for 50 epochs. Optimisation is performed with AdamW (weight-decay $1 \times 10^{-4}$).

Hyper-parameters are selected automatically with *Optuna* v4.4 (2000 trials, Tree-Parzen Estima-tor). The search space is LR $\sim \log U(10^{-5}, 10^{-3})$, $\lambda_{1,2} \sim U(0.01, 0.20)$, and $k \in \{1, \ldots, 9\}$. For UCF-Crime, the best trial yields: learning-rate $5.69 \times 10^{-4}$, Category classifier LR $1.29 \times 10^{-3}$, $k{=}8$, $\lambda_1{=}0.0999$, $\lambda_2{=}0.0121$, and batch-size 16. For XD-Violence, the optimal settings are: learning-rate $3.10 \times 10^{-4}$, Category classifier LR $1.10 \times 10^{-3}$, $k{=}7$, $\lambda_1{=}0.08$, $\lambda_2{=}0.01$, and batch-size 12. These configurations are re-trained on the full training set to produce the results reported in the paper.

**InternVideo 2.5 Caption Generation.**  Each 64-frame snippet is fed to the InternVideo 2.5 Chat-8B model to obtain three clue-specific captions (*Action*, *Environment*, *Object*). We adopt the inference configuration {do_sample=False, max_new_tokens=30, num_beams=1}, thereby producing a single deterministic sentence per clue that summarises the entire 64-frame tem-poral context.

**(a) Action**

Only describe what the **person or people are doing in the scene**.
Do not describe the place, objects, or appearance.
Use a single short sentence that starts with a verb and follows this template:

Template: **"{subject} is {action}"**

Examples:
- "A man is running."
- "Two people are fighting."
- "A woman is walking alone."

**(b) Environment**

Describe the **physical environment or setting of this scene**.
Do not describe people or objects. Focus only on the place, lighting, or spatial layout.
Use a single short sentence that follows this template:

Template: **"The scene takes place in {location}."**

Examples:
- "The scene takes place in a small grocery store."
- "The scene takes place on a quiet suburban street."
- "The scene takes place in a dimly lit hallway."

**(c) Object**

Describe **only the most notable object or item in the scene**.
Focus on objects that stand out, are being interacted with, or are out of place.
Use a single short sentence that follows this template:

Template: **"The most notable object is {object}."**

Examples:
- "The most notable object is a black handgun."
- "The most notable object is a broken window."
- "The most notable object is a backpack on the floor."

Figure S1: Prompt templates for CLUE–specific captioning: We use InternVideo 2.5 prompt separately with three tightly constrained, semantic queries: (a) **Action**, (b) **Environment**, and (c) **Object**.

**CLIP Text-Encoder Embedding.** We encode every generated caption with the CLIP text backbone ViT/H-14. The snippet-level text embedding $e_{\text{text}} \in \mathbb{R}^{T \times d_{\text{text}}}$, where $d_{\text{text}}=1024$, is obtained by taking the token of the final transformer block and projecting it to the shared fusion space of dimension $d_{\text{proj}}$.

**CLIP Vision-Encoder Embedding.** Concurrently, a single key-frame of each snippet is fed to the CLIP vision backbone ViT/H-14. The pooled image representation $e_{\text{vis}} \in \mathbb{R}^{T \times d_{\text{vis}}}$, with $d_{\text{vis}}=1024$, is again linearly mapped to the common dimensional space to facilitate modality fusion.

## 3. CLUE-VAD BENCHMARK

**InternVideo 2.5 Prompt Template.** To ensure structured and semantically consistent caption generation for the CLUE-VAD Benchmark, we carefully design three distinct, constrained prompt templates as shown in Figure. S1. Each prompt targets one of the three core semantic roles: *Action*, *Environment*, and *Object*. Specifically, the **Action** prompt strictly instructs the model to produce a concise, verb-driven sentence depicting solely the activities performed by individuals within the scene. It explicitly prohibits mentioning environmental contexts, appearance details, or involved objects, enforcing a clear template format: "{*subject*} *is* {*action*}". Example generations include "*Two people are fighting*" or "*A woman is walking alone*," ensuring consistent linguistic alignment to human action semantics.

The **Environment** prompt focuses exclusively on capturing spatial and contextual aspects of the scene. It directs the model to generate a short descriptive sentence beginning with the phrase "*The*

> Action: "Extract one **action-related keyword (verb)** from the following sentence. Only output the keyword."
>
> Object: "Extract one **main object-related keyword (noun)** from the following sentence. Only output the keyword."
>
> Environment: "Extract one **keyword related to environment** from the following sentence. Only output the keyword."

Figure S2: Prompt templates for Keyword Finding: We use Llama 3-8B model to search separately three keyword: (a) **Action**, (b) **Environment**, and (c) **Object**.

*scene takes place in* {*location*}". Here, model is explicitly restricted from referencing people, actions, or individual objects, thus isolating the environmental context. Representative outputs include *"The scene takes place in a quiet suburban street"* or *"The scene takes place in a dimly lit hallway,"* effectively providing contextual settings that complement anomaly interpretation.

Lastly, the **Object** prompt targets the identification and description of the most salient or contextually abnormal object present in the scene. The template, *"The most notable object is* {*object*},*"* guides model to succinctly identify standout items that may signal anomaly indicators or deviations. Examples such as *"The most notable object is a broken window"* or *"The most notable object is a black handgun"* illustrate the model's capability to highlight meaningful objects crucial for fine-grained anomaly reasoning.

**CLUE-specific Generated Caption Examples.** InternVideo 2.5, guided by the CLUE-constrained prompts of Figure S1, produces captions that are remarkably sensitive to the temporal evolution of an event (Figure S3). *Arrest030.mp4* the Action caption shifts from the innocuous *"A man is tampering with the ATM."*—a clear precursor that signals suspicious intent—to *"Two police officers are arresting a suspect.",* precisely describing the anomalous climax. The accompanying Environment sentence narrows from *"The scene takes place in an ATM area."* to *"The scene takes place in a dimly lit ATM area.",* reflecting the intensified focus. The Object evolves from *"an ATM machine being opened."* to *"a police handgun.",* highlighting the appearance of force. A similar three-stage narration is observed in *Explosion029.mp4*: the fueling activity is first observed (precursor), followed by the ignition event, and finally the salient object switches from *fuel pump* to *a fire coming from the car*.

Across all cases, the model exhibits a consistent ability to (1) anticipate abnormality by describing subtle preparatory actions (*"A man is lighting the Christmas tree"*), (2) accurately depict the disruptive moment itself (*"A fire is breaking out near the gas tank"*), and (3) capture immediate post-event consequences, such as objects in flames or victims on the ground. This temporal sensitivity, coupled with strict role isolation, provides CLUE-VAD with rich, structured evidence that improves both anomaly localization and the interpretive value of its explanations.

**Llama 3.8 Prompt Template.** To further condense the caption text into a single, highly informative token that can be used for lightweight statistical analysis, we query the model with three CLUE-aware keyword prompts in Figure S2. Each prompt is issued independently so that the model focuses on exactly one semantic dimension at a time—**(a) Action**, **(b) Environment**, or **(c) Object**. The prompts are intentionally minimal (see below) and the generation configuration is deterministic ensuring that the model returns only one keyword. All prompts are issued through the same inference pipeline with the model checkpoint(Meta-Llama-3-8B-Instruct).

**Analysis of CLUE Keyword Frequency.** TableS1 shows a pronounced semantic polarity between normal and abnormal segments in the clue of Action. Routine clips are saturated with low-energy locomotion verbs such as *walking*, *standing*, *talking*, that signal pedestrian passivity. whereas their anomalous counterparts pivot to high-energy or goal-directed actions that typify each crime: *fighting* and *holding* in *Assault*, *pushing* and *taking* in *Robbery*, and *burning* or *bending* (as a pre-ignition cue) in *Arson*. Several categories also exhibit *anticipatory* verbs that flag the onset

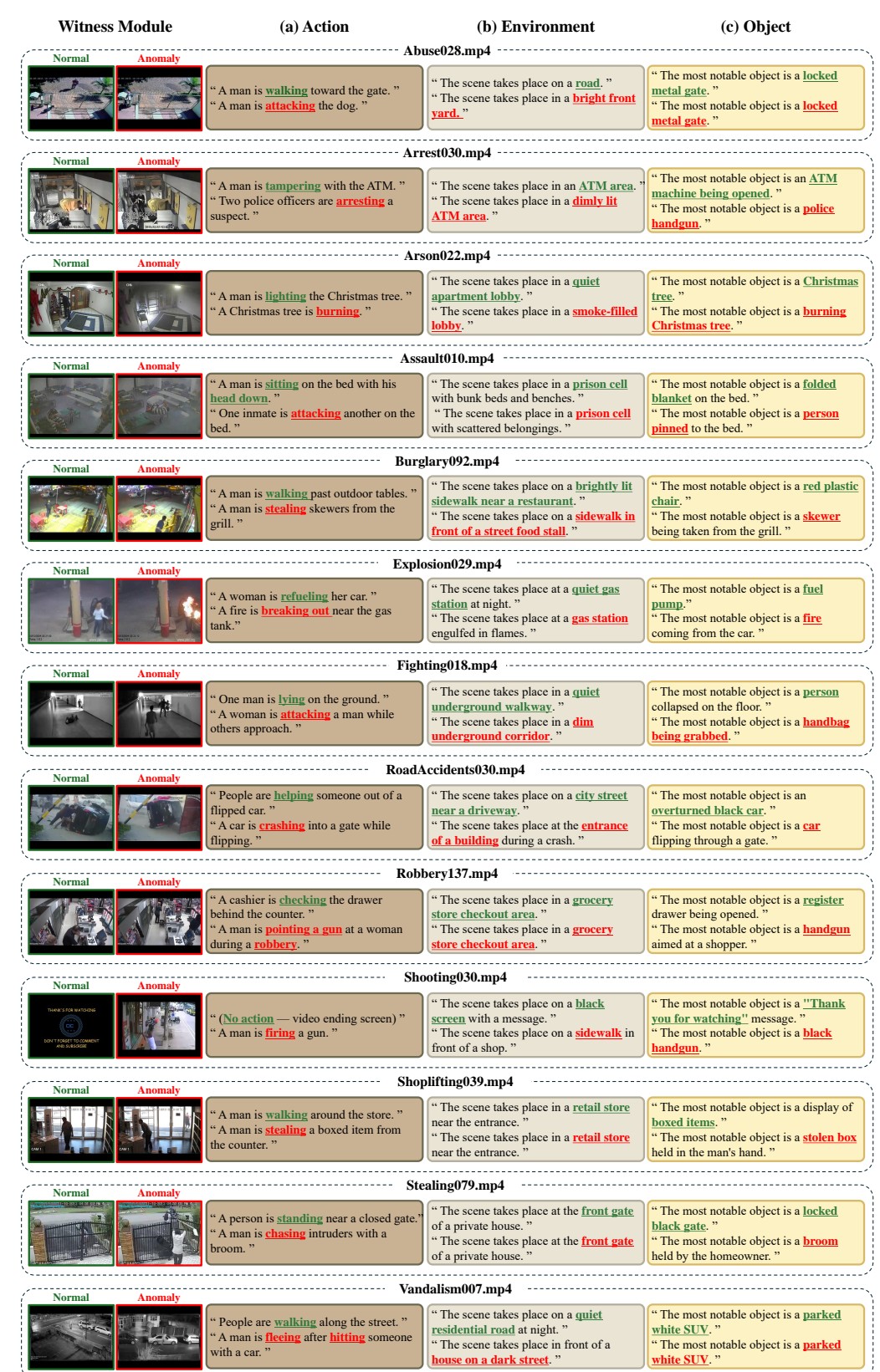

Figure S3: Representative caption examples for each semantic role in the CLUE-VAD Benchmark: (a) **Action**, (b) **Environment**, and (c) **Object**. Each caption captures specific semantic elements that facilitate fine-grained interpretation of anomalous events.

| Category | Normal (Top–5) | Anomaly (Top–5) |
|---|---|---|
| Abuse | walking(111), doing(17), trying(16), standing(8), riding(6) | walking(6), **attacking(3)**, riding(2) |
| Arrest | standing(335), walking(320), talking(204), sitting(72), moving(72) | walking(108), standing(72), **arresting(56)**, running(42), **lying(32)** |
| Arson | walking(288), standing(166), moving(137), doing(136), are(79) | walking(150), moving(68), running(66), **burning(36)**, bending(31) |
| Assault | walking(388), sitting(234), sweeping(108), standing(72), **lying(64)** | **fighting(170)**, talking(116), standing(65), walking(32), holding(28) |
| Burglary | walking(1 351), are(649), doing(337), moving(250), driving(168) | walking(248), standing(133), sitting(120), cleaning(53), dancing(42) |
| Explosion | walking(1 271), standing(308), dancing(271), running(192), talking(163) | walking(153), standing(63), running(48), are(34), is(28) |
| Fighting | talking(103), walking(74), standing(71), **lying(43)**, dancing(28) | **fighting(83)**, walking(48), helping(20), trying(16), running(15) |
| RoadAccidents | walking(372), riding(245), crossing(242), standing(94), **lying(81)** | walking(35), riding(23), **lying(22)**, crossing(18), moving(13) |
| Robbery | walking(54), standing(45), talking(40), doing(25), riding(20) | **pushing(34)**, taking(32), walking(31), standing(22), **fighting(20)** |
| Shooting | walking(1 432), standing(806), talking(227), lying(176), sitting(156) | standing(93), walking(85), **fighting(66)**, **lying(55)**, running(33) |
| Shoplifting | standing(830), walking(771), talking(763), looking(668), sitting(281) | looking(97), standing(78), counting(76), talking(49), playing(40) |
| Stealing | standing(214), walking(187), talking(84), riding(69), is(62) | standing(194), trying(36), walking(33), riding(27), opening(18) |
| Vandalism | walking(238), moving(81), standing(65), doing(37), trying(36) | walking(52), trying(31), cleaning(12), running(8), standing(8) |

Table S1: Action CLUE keywords: Top-5 verbs per anomaly category for normal and anomalous.

| Category | Normal (Top–5) | Anomaly (Top–5) |
|---|---|---|
| Abuse | parking(76), suburban(73), road(18), area(5), environment(2) | parking(5), road(4), street(2) |
| Arrest | room(639), hallway(295), lobby(202), **office(135)**, bank(44) | room(222), hallway(73), street(70), road(40), **office(23)** |
| Arson | hallway(355), room(306), parking(149), street(62), bus(62) | porch(165), hallway(97), bus(51), room(48), street(31) |
| Assault | room(436), **prison(388)**, dormitory(191), alley(54), parking(27) | room(336), store(104), shop(28), parking(25), **prison(24)** |
| Burglary | porch(866), parking(770), **office(717)**, room(369), store(235) | porch(202), parking(184), room(163), **office(154)**, hallway(89) |
| Explosion | alley(482), parking(390), area(286), courtyard(285), station(239) | **industrial(111)**, station(55), forest(51), street(31), city(26) |
| Fighting | garage(176), subway(113), hallway(97), lobby(70), background(15) | garage(95), subway(81), hallway(41), building(31), lobby(15) |
| RoadAccidents | street(521), intersection(257), city(219), road(142), suburban(84) | intersection(32), street(25), city(23), suburban(20), road(17) |
| Robbery | parking(107), **store(77)**, driveway(61), backyard(30), shop(4) | **store(145)**, driveway(80), backyard(10), parking(9) |
| Shooting | parking(1 600), street(369), room(363), **bank(269)**, office(254) | driveway(158), street(125), suburban(66), room(46), store(44) |
| Shoplifting | **store(3204)**, shop(430), office(305), room(257), background(45) | store(280), room(165), shop(38), office(11), bank(1) |
| Stealing | suburban(227), parking(191), driveway(146), residential(102), street(83) | suburban(94), driveway(92), parking(74), street(65), lot(26) |
| Vandalism | parking(257), store(173), sidewalk(44), hallway(41), suburban(13) | parking(40), store(33), sidewalk(28), suburban(14), outside(8) |

Table S2: Environment CLUE keywords: Top-5 location descriptors per category for normal and anomalous.

| Category | Normal (Top–5) | Anomaly (Top–5) |
|---|---|---|
| Abuse | van(75), gate(49), dog(16), cat(11), bag(4) | gate(6), umbrella(3) |
| Arrest | **handgun(464)**, bag(148), chair(94), car(91), monitor(87) | **handgun(208)**, car(132), tarp(54), whiteboard(12), chair(6) |
| Arson | tree(458), van(193), chair(100), car(71), door(56) | tree(228), van(40), car(21), sign(16), cat(10) |
| Assault | mop(167), blanket(100), broom(72), car(64), bucket(61) | bag(32), table(24), box(20), shirt(17), **handgun(16)** |
| Burglary | car(1 327), television(259), door(180), monitor(167), mat(158) | car(387), monitor(112), door(78), box(55), television(42) |
| Explosion | car(681), **pump(213)**, vehicle(210), tree(204), truck(121) | car(98), building(26), motorcycle(21), **pump(14)**, **torch(12)** |
| Fighting | **handgun(95)**, bag(58), backpack(44), tire(24), chair(21) | **handgun(56)**, bag(42), map(24), whiteboard(20), chair(18) |
| RoadAccidents | car(844), truck(251), motorcycle(82), van(69), bus(44) | car(111), truck(22), van(10), motorcycle(7), bus(4) |
| Robbery | car(152), door(46), **handgun(22)**, truck(15), motorcycle(7) | car(82), **handgun(56)**, bag(48), door(31), truck(12) |
| Shooting | car(1 342), **handgun(324)**, sign(269), bag(183), umbrella(154) | car(245), **handgun(122)**, door(25), bag(20), bucket(16) |
| Shoplifting | **handgun(601)**, bag(590), monitor(424), box(355), chair(205) | **handgun(84)**, bag(83), box(82), laptop(31), wallet(28) |
| Stealing | car(510), gate(162), motorcycle(76), van(40), door(8) | car(198), gate(55), van(42), motorcycle(41), door(25) |
| Vandalism | car(266), door(146), refrigerator(68), window(14), glass(12) | car(43), door(42), refrigerator(23), dog(8), window(5) |

Table S3: Object CLUE keywords: Top-5 salient objects per category for normal and anomalous.

of wrongdoing—*opening* ahead of forced entry in *Stealing* or *running* before confrontation in *Arrest*—indicating that the role-aware caption pipeline captures temporal precursors as well as the peak anomalous motion, while suppressing background behaviours that would otherwise dilute the signal.

TableS2 shows that Normal footage is largely situated in public areas like *parking* lots, *room* interiors, suburban *street*, where background motion is expected. When anomalous events unfold, these spatial descriptors tighten around crime-relevant settings: *Explosion* shifts from open *courtyard* or *station* references to high-risk *industrial* compounds and forested storage areas. *Burglary* moves from *office* and *store* exteriors to private *porch* and interior *room* locations, and traffic accidents transition from generic *street* mentions to specific *intersection* hotspots. Notably, *Shooting* anomalies emphasise residential *driveway* and suburban *street* contexts—areas where the sudden appearance of a firearm is most disruptive—while *Shoplifting* anomalies concentrate almost exclusively on *store* interiors. These environment-specific shifts demonstrate that our role-aware captioning not only recognises the objects and actions that constitute an anomaly but also grounds them in the precise spatial context where their abnormality becomes evident.

TableS3 shows a clear shift in Object terms from benign everyday items in normal clips to crime-specific artefacts in anomalous ones. In non-anomalous segments, vehicles and furniture dominate (*car*, *bag*, *television*), mirroring the prevalence of public parking lots, indoor stores, and office spaces in UCF-Crime. Once an anomaly occurs, however, the distributions pivot toward threat indicators: weapons such as *handgun* surge in *Arrest*, *Shooting*, and *Robbery*; ignition sources like *torch* and fuel-station *pump* emerge in *Explosion*; and easily removed goods (*wallet*, *laptop*, *box*) become salient in *Shoplifting* and *Burglary*. The appearance of makeshift tools (*tarp*, *whiteboard*) in anomalous *Arrest* scenes, or *gate* and *door* during *Stealing*, further confirms that the CLUE pipeline isolates tangible cues that are directly implicated in the criminal act.

**Structured Output Representation in JSON.** Each video processed by CLUE-VAD is represented through structured JSON files named {*video_name*}_{*CLUE*}.*json*, where each segment (defined as a sequence of 16 consecutive frames) is indexed numerically as a unique key. For each segment, we record detailed frame-boundary information (start_frame, end_frame)

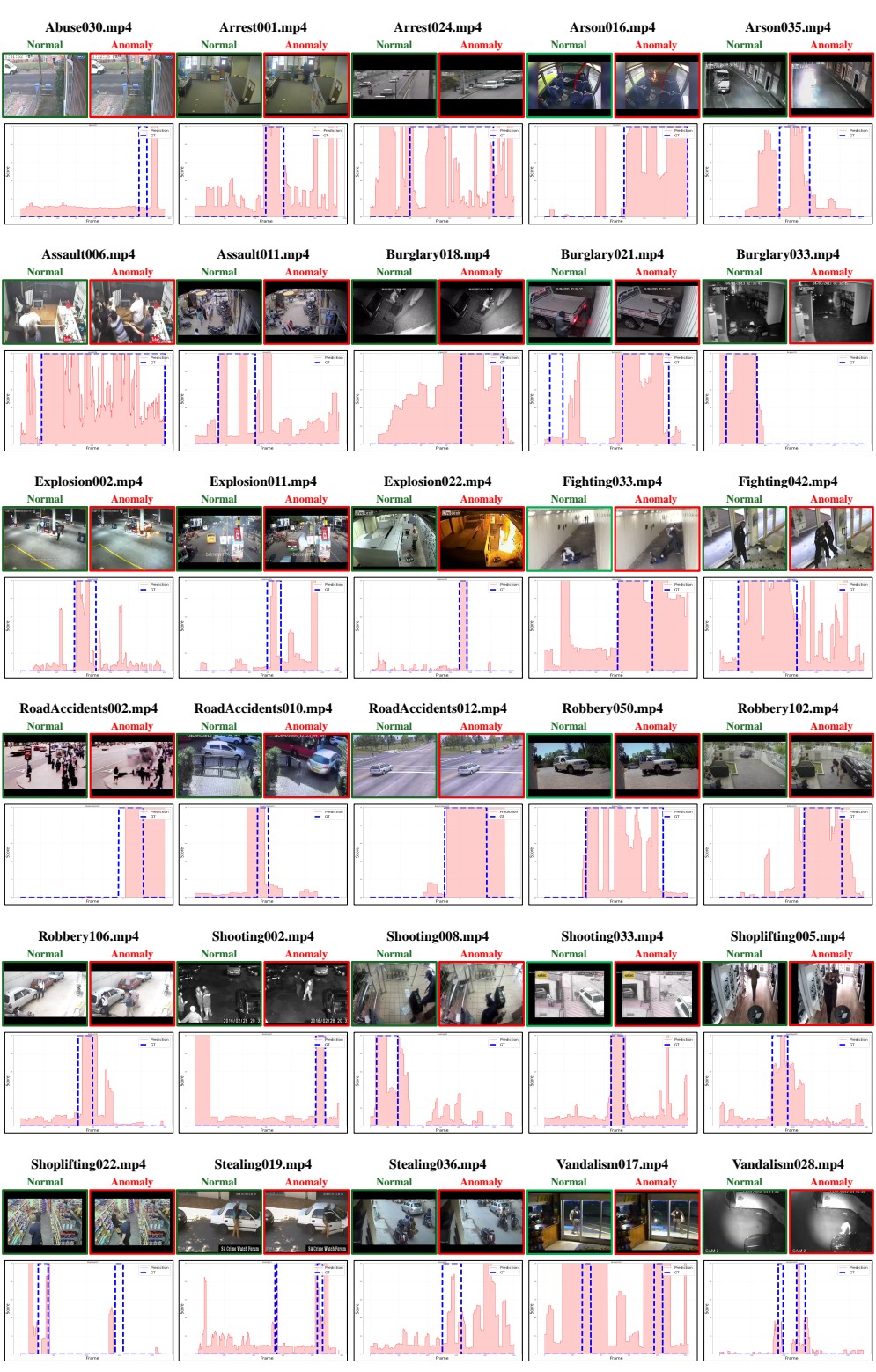

Figure S4: Frame-level anomaly prediction results across all anomaly categories in UCF-Crime: for each representative video, we display frames from normal and anomalous segments along with their corresponding predicted anomaly scores (red shading). Ground truth anomaly intervals are indicated by the dashed blue lines, highlighting the model's ability to effectively capture anomalies across diverse scenarios and temporal contexts.

alongside descriptive semantic annotations. Specifically, each segment is annotated with a natural-language `caption` generated by InternVideo 2.5, accompanied by a single, semantically condensed `keyword` extracted by the Llama 3-8B model. Additionally, each segment contains critical internal model metrics computed during inference: `attn`, `deviation`, and `score_weight`. Here, the `attn` vector quantifies the relative attention weights assigned to each semantic clue (Action, Object, Environment) by the Category-Aware Fusion module, the `deviation` vector reflects the Z-score normalized deviation of each role from their respective temporal means, and the `score_weight` indicates each role's contribution to the final segment-level anomaly prediction.

This structured and integrated JSON file format significantly enhances interpretability and facilitates deeper contextual understanding of anomalous events. By clearly separating and annotating semantic roles within each segment, researchers can rapidly pinpoint the precise context and cause underlying an anomaly, greatly simplifying follow-up analysis or statistical evaluations. Moreover, the recorded internal metrics—`attn`, `deviation`, and `score_weight`—provide valuable insights into the model's decision-making process, enabling more granular investigation into the temporal evolution of anomalies and their underlying semantic triggers. Ultimately, this unified and structured JSON representation promotes reproducibility, analytical transparency, and efficient advancement of future research in video anomaly detection.

## 4. CLUE-VAD BASELINE

**Analysis of Frame-level Anomaly Predictions.** Figure S4 further demonstrates that CLUE-VAD is not only reactive to the peak of an abnormal episode but is also proactive in signalling the contextual build-up that precedes it and the residual aftermath that follows. In *Arson035.mp4*, for example, the anomaly curve begins to climb during the actor's suspicious bending motion several seconds before visible flames appear. A comparable early rise is seen in *Burglary033.mp4*, where the score increases while the intruder is still tampering with the lock. Conversely, in *Explosion011.mp4* the model maintains elevated scores even after the initial blast, capturing lingering smoke and crowd panic, evidence that the detector integrates post-event cues rather than collapsing to a single impulse. This temporal sensitivity stems from the role-aware caption pipeline: precursor verbs such as *bending*, *opening*, or *tampering* and aftermath objects such as *fire*, *smoke*, or *collapsed car* provide the latent tokens that nudge the anomaly score upward before and after the ground-truth window.

Detecting these pre- and post-event signatures is crucial in real-world surveillance settings, where early warnings can mitigate damage and persistent monitoring can aid forensics. By disentangling *Action*, *Object*, and *Environment* clues, CLUE-VAD learns to associate subtle contextual shifts—dim lighting in a closed ATM lobby or the sudden appearance of a fuel pump—with imminent risk, while suppressing routine background motion such as pedestrians *walking* or employees *standing*. The result is a temporally smooth yet anticipatory prediction profile that respects the narrative structure of an incident: calm, escalation, climax, and aftermath. Such contextual understanding underpins the system's low false-positive rate on benign footage (green segments remain near zero) and its high localization accuracy across diverse anomaly types, validating the importance of fine-grained, clue-guided modeling for frame-level anomaly detection.

**Category-wise Performance Analysis.** Table S4 indicates that CLUE-VAD attains uniformly strong AUC scores for violence-oriented crimes such as *Assault* (0.81), *Robbery* (0.86), *Stealing* (0.86), and *Vandalism* (0.86). These categories typically contain distinctive high-impact actions (*fighting*, *taking*) and salient objects (*handgun*, *gate*), allowing the role-aware fusion module to isolate abnormal segments with high confidence. Burglary and Fighting also benefit from consistent scene layouts (*porch*, *garage*) that the Environment clue captures effectively, yielding mid-to-high AUC values around 0.75.

In contrast, categories such as *Explosion* (0.51) and *Abuse* (0.59) exhibit lower scores, largely due to ambiguous

| Category | AUC (%) |
|---|---|
| Abuse | 58.87 |
| Arrest | 61.72 |
| Arson | 64.98 |
| Assault | 80.97 |
| Burglary | 75.79 |
| Explosion | 51.35 |
| Fighting | 75.18 |
| RoadAccidents | 62.52 |
| Robbery | 86.27 |
| Shooting | 71.09 |
| Shoplifting | 61.84 |
| Stealing | 85.96 |
| Vandalism | 85.80 |

Table S4: Category-wise AUC scores on the UCF-Crime.

pre-event motion and the scarcity of discriminative objects in many clips. *RoadAccidents* (0.63) and *Shoplifting* (0.62) show moderate performance, both suffer from subtle or gradual deviations that require long-term temporal reasoning. These observations highlight two directions for future work: (i) enriching the Environment cue with finer scene-change detectors to better capture sudden global disruptions (e.g. flashes, fire), and (ii) integrating longer temporal windows to enhance sensitivity to gradual behavioural shifts in theft-like scenarios. Overall, the category-wise analysis confirms that disentangling semantic roles enables robust detection in action– and object-centric crimes, while revealing the remaining challenges for events driven primarily by environmental cues or slowly evolving contexts.

| Rank | Category Classifier ACC (%) | UCFC AUC (%) |
|------|------------------------------|--------------|
| 1 | 74.23 | 88.12 |
| 2 | 83.45 | 88.74 |
| 3 | GT | 89.96 |

Table S5: Effect of category classifier accuracy on UCFC anomaly detection performance.

**Details of the Category Classifier.** The proposed Category-Aware CLUE Weighting relies on selecting the appropriate clue weights $\{w_{\text{Act}}^{(c)}, w_{\text{Env}}^{(c)}, w_{\text{Obj}}^{(c)}\}$ for each anomaly category $c$. To achieve this, we employ a category classifier that predicts the category $\hat{c}$ of each segment based on its CLUE-wise embeddings. This prediction is then used to index the Case Log and retrieve the category-specific clue weights. The classifier is implemented as a two-layer MLP with ReLU activation, taking as input the concatenated CLUE embeddings:

$$x_t = \big[\, c_{\text{Act}}^{(t)} \,\|\, c_{\text{Env}}^{(t)} \,\|\, c_{\text{Obj}}^{(t)} \,\big],$$

and producing class probabilities via

$$p(c \mid x_t) = \text{softmax}(W_2 \cdot \sigma(W_1 x_t + b_1)).$$

During training, the ground-truth category $c^*$ is used to supervise the classifier using Cross-Entropy loss, and simultaneously update the Case Log Embedding for each category. During inference, however, only the predicted category $\hat{c}$ is available; hence it is used to retrieve the learned clue weights from the Case Log, which subsequently guide the category-aware fusion. As a result, the accuracy of the classifier directly affects the final anomaly-scoring performance.

To quantify this relationship, we report the final UCFC AUC as a function of classifier accuracy. Table S5 summarizes the results. Increasing the classifier accuracy from 74.23% to 83.45% yields a performance gain from 88.12 to 88.74 AUC. Moreover, using oracle ground-truth categories leads to an upper bound of 89.96 AUC. These findings confirm that the category classifier indeed influences the end-to-end anomaly scoring, and motivate the importance of accurate category estimation in category-aware CLUE fusion.

**CLIP Backbone Analysis.** Table S6 reports the effect of different CLIP encoders on the performance of CLUE-VAD. Across four commonly used CLIP backbones—ViT-B/16, ViT-B/32, ViT-L/14, and ViT-H/14—the model exhibits only minor variation in UCF-Crime AUC (88.32 $\rightarrow$ 88.74). The performance differences remain within a narrow 0.4% range, indicating that CLUE-VAD is not overly sensitive to backbone capacity. Moving from ViT-B to ViT-L yields a modest improvement

| CLIP backbone | UCF-Crime (AUC(%)) |
|---------------|---------------------|
| ViT-B/16 | 88.32 |
| ViT-B/32 | 88.36 |
| ViT-L/14 | 88.43 |
| ViT-H/14 | 88.74 |

Table S6: Performance of CLUE-VAD under different CLIP backbones.

(+0.11~+0.31), while the largest-scale backbone, ViT-H/14, provides only a slight additional gain over ViT-L/14 (+0.31). These trends suggest that the majority of the performance comes from the structured CLUE decomposition and category-aware fusion rather than the representational power of larger CLIP encoders.

Overall, the backbone sweep demonstrates that CLUE-VAD is robust across diverse CLIP variants, and that even lightweight backbones such as ViT-B/16 and ViT-B/32 already offer strong results.

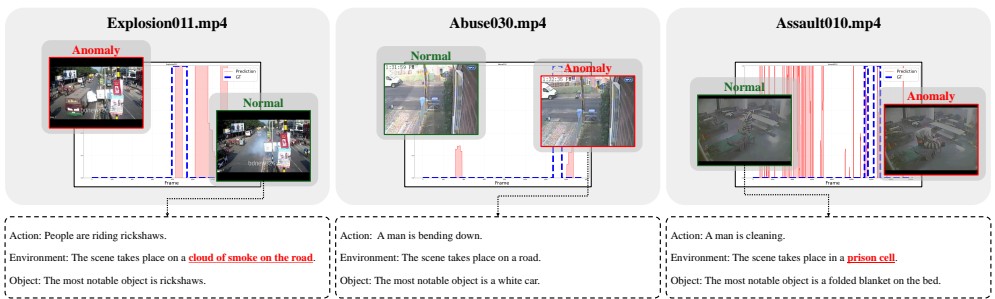

Figure S5: **Failure cases of CLUE-VAD where individual CLUEs appear normal but their combination yields contextually abnormal interpretations.** (Left) In *Explosion011.mp4*, residual smoke remains after the explosion, causing the Environment CLUE to trigger a high anomaly score despite normal actions and objects. (Middle) In *Abuse030.mp4*, the anomalous interaction is extremely small and not captured by the captioner, producing benign-looking CLUEs and resulting in missed detection. (Right) In *Assault010.mp4*, although the actions and objects correspond to a normal cleaning scene, the prison-cell environment induces a strong scene prior that elevates anomaly scores across the segment. These cases highlight limitations arising from post-event artifacts, visually subtle anomalies, and biased or rare environments.

This confirms that the framework does not depend on a particular pretrained backbone and that improvements over prior explainable VAD methods stem from the proposed CLUE reasoning rather than from scaling encoder size.

**Failure Case Analysis.** Figure S5 illustrates representative failure modes of CLUE-VAD where each individual CLUE appears locally benign, yet their combination is contextually abnormal. Although CLUE-VAD provides fine-grained, clue-wise reasoning, the model still exhibits notable failure modes where each semantic clue appears individually normal, yet their combination implicitly indicates an abnormal situation.

In Explosion011.mp4, the anomalous event (explosion) occurs before the captured segment, leaving only residual smoke in the scene. While the Action ("people riding rickshaws") and Object ("rickshaw") CLUEs are perfectly normal, the Environment CLUE highlights "a cloud of smoke on the road," causing the model to assign an elevated anomaly score. Since CLUE-VAD treats smoke as a strong environmental prior for Explosion events, post-event remnants trigger false alarms even when the actual action is mundane.

In Abuse030.mp4, the true anomalous interaction occupies only a tiny region of the frame. Although Environment and Object remain normal, the Action CLUE ("a man is bending down") fails to capture the subtle abusive contact. This reveals a limitation of CLUE-based reasoning: if the captioning model detects only the dominant scene context, minor but critical interactions (e.g., slight physical aggression) are not reflected in any CLUE, causing under-detection.

In Assault010.mp4, the scene unfolds inside a prison cell, a location rarely associated with normal behavior in the training distribution. Despite the Action ("a man is cleaning") and Object ("a folded blanket on the bed") being innocuous, the Environment CLUE strongly emphasizes prison-like elements, which the model tends to associate with abnormal categories such as Assault or Abuse. As a result, anomaly scores remain globally high even during normal segments, suggesting that domain-specific environments introduce a structural prior that dominates CLUE fusion.

These failure cases reveal that CLUE-VAD, while effective at disentangling semantic cues, can misinterpret (i) persistent environmental artifacts, (ii) visually subtle anomalous actions, and (iii) biased or rare environments. Importantly, in all three cases, each individual CLUE appears normal, but their real-world co-occurrence is contextually abnormal—highlighting a key challenge in compositional anomaly reasoning and motivating future work on joint CLUE consistency modeling and environment-agnostic calibration.

**Effect of Captioning Models.** To directly evaluate the dependence of CLUE-VAD on the captioning module, we conducted a captioner-swap experiment in which

| VLM Captioning model | UCF-Crime (AUC(%)) | XD (AP(%)) |
|---|---|---|
| BLIP-2 | 86.46 | 86.13 |
| Video-LLaVA | 88.68 | 86.92 |
| CLUE-VAD (ours) | 89.23 | 87.64 |

Table S7: Captioner-swap evaluation under the CLUE-VAD pipeline.

the Witness module was paired with several off-the-shelf video–language captioners of differing architectures and training corpora. As shown in Table S7, multimodal CLUE-VAD remains stable across captioners: on UCF-Crime, the model achieves 86.46% AUC with BLIP-2 captions, 88.68% with Video-LLaVA captions, and 89.23% with our default captioner. On XD-Violence, the corresponding AP scores are 86.13%, 86.92%, and 87.64%. All variants lie within a narrow range (roughly 3 AUC points on UCF-Crime and 1.5 AP points on XD-Violence), and none exhibit performance collapse. These results indicate that, as long as the captioner reasonably identifies the main actors, salient objects, and scene context, the CLUE decomposition and category-aware fusion can recover stable anomaly scores regardless of the specific captioner used.

