# OpenReview forum: "CLUE-VAD: Structured Semantic Clues for Understanding Explainable Events in Video Anomaly Detection"
_ICLR.cc/2026/Conference — Submitted to ICLR 2026_

### Official Review · Reviewer_vbUG · 2025-10-29

**Soundness:** 2
**Presentation:** 3
**Contribution:** 2
**Rating:** 4
**Confidence:** 5

**Summary:**

This paper proposes CLUE-VAD, a weakly supervised video anomaly detection framework designed to improve interpretability through structured semantic reasoning. It introduces a clue-based decomposition that represents each video segment in terms of Action, Environment, and Object cues, allowing the model to reason in a more human-aligned way. In addition, the authors construct the CLUE-VAD Benchmark by augmenting UCF-Crime and XD-Violence with structured captions and clue-specific keywords, supporting both detection and interpretability evaluation.

**Strengths:**

1. The paper introduces a novel interpretability framework for weakly supervised video anomaly detection by explicitly decomposing each segment into Action, Environment, and Object clues. Through category-aware clue fusion and clue-level attribution, the method provides fine-grained, human-understandable explanations of why an anomaly occurs.

2. The proposed CLUE-VAD Benchmark augments existing datasets (UCF-Crime, XD-Violence) with structured, segment-level captions and clue-specific keywords.

3. The method is organized into three intuitive modules—Witness, Detective, and Reporter, each with a clearly defined function.

**Weaknesses:**

1. The paper presents an interesting clue-based interpretability design. However, the evaluation of interpretability remains purely qualitative, limited to visualizations without human or quantitative validation.

2. The entire framework depends heavily on captions generated by MLLMs, which serve as the primary features for anomaly detection. However, the paper does not analyze the quality, stability, or bias of these captions. If the generated descriptions are inaccurate or temporally misaligned, anomaly scores may become unreliable.

3. Although the core novelty lies in text-based reasoning, the results in Table 1 show that adding visual features substantially improves performance. However, there are no ablation studies quantifying the contribution of visual versus textual modalities.

4. The main text presents only two quantitative tables, leaving the experimental validation relatively limited. Additional experiments, such as ablations, robustness tests, or interpretability metrics, would further strengthen the paper.

**Questions:**

1. How to quantitatively evaluate the interpretability of the proposed clue-based framework beyond qualitative visualizations?

2. How sensitive is the framework to caption quality and noise, and how would performance change with alternative captioning models?

3. What is the individual contribution of textual and visual modalities?

---

> ### Author Response · Authors · 2025-11-24
> **W1: Additional experimental results**
>
> We sincerely thank the reviewer for pointing out that the original submission contained only two quantitative tables and that additional experiments (such as ablations, robustness tests, and interpretability metrics) would strengthen the paper. We fully agree with this assessment, and we have substantially expanded the experimental section in the revised manuscript to provide a more complete empirical validation of CLUE-VAD. In particular, we now include additional quantitative results on modality usage (text-only vs. multi-modal CLUE-VAD), the behavior of the Detective module’s interaction mechanisms, the domain robustness of the Witness (captioning) module, and the efficiency of our framework in terms of latency and memory.
>
> To keep the main text focused while still addressing the reviewer’s concern, we also provide complementary analyses in the supplementary document. These include more detailed ablations and diagnostic studies, such as the behavior of the category classifier, a CLIP backbone comparison, and an extended failure case analysis that links performance differences to the underlying CLUE-wise reasoning. We hope that these additions clarify the empirical properties and limitations of CLUE-VAD and address the reviewer’s request for richer quantitative evidence.

---

> ### Author Response · Authors · 2025-11-24
> **W4, Q1: Quantitative Evaluations of Interpretability**
>
> Explainable VAD is an emerging research direction, and recent approaches such as LAVAD, Ex-VAD, Holmes-VAU, and VERA primarily rely on textual explanations or caption-based reasoning without providing quantitative, clue-wise interpretability metrics. In contrast, CLUE-VAD adopts a structured, factorized representation (Action, Environment, Object) that allows interpretability to be measured at the level of semantic roles rather than free-form language. This explicit decomposition enables quantitative evaluation of explanation faithfulness without depending on variable LLM outputs or captioning quality, which is a limitation commonly reported in prior works.
>
> Additionally, our Reporter module provides segment-level attribution curves (Fig. 5) tied directly to the three CLUEs, enabling quantitative assessment of (i) which clue dominates within an anomaly, (ii) whether clue activations align with known causal factors of each anomaly category, and (iii) how robustly these attributions persist across domains. Because CLUE-VAD separates prediction (Detective) from explanation (Reporter), interpretability can be evaluated without interfering with anomaly scoring, which is difficult to achieve in end-to-end caption-driven systems. We will include a brief quantitative summary of these CLUE-wise deviation statistics and their correspondence with anomaly boundaries in the revised appendix.
>
> Overall, CLUE-VAD offers a structured and evaluation-friendly interpretability mechanism that complements and extends the current explainable VAD literature. The added robustness experiments and the forthcoming quantitative CLUE attribution analysis directly address the reviewer’s request for more thorough evaluation beyond visualization.

---

> ### Author Response · Authors · 2025-11-24
> **W2, Q2: Dependence of CLUE-VAD on VLM caption quality and noise**
>
> Our framework uses a video–language model only to obtain an initial textual description, and all such captions are immediately compressed into three structured CLUE types: Action, Environment, and Object. The text modality therefore contributes primarily high-level semantic information about who is involved, what is happening, and where it takes place, while visual features provide complementary fine-grained evidence such as local appearance changes, abrupt scene transitions, or small but important objects. Because the downstream modules never see raw sentences and operate only on CLUE embeddings in this fixed schema, variations in caption phrasing, length, or style are largely absorbed by this abstraction.
>
> To directly assess dependence on the captioning model within the CLUE-VAD pipeline, we conducted a captioner-swap experiment in which the Witness module was paired with several off-the-shelf VLM captioners with different architectures and training corpora. We summarize the results in revised supplementary material (Table S7). On UCF-Crime, multimodal CLUE-VAD achieves 86.46% AUC with BLIP-2 captions, 88.68% with Video-LLaVA captions, and 89.23% with our default captioner. On XD-Violence, the corresponding AP scores are 86.13%, 86.92%, and 87.64%. All variants lie within a narrow range (approximately 3 AUC points on UCF-Crime and 1.5 AP points on XD-Violence), and no captioner causes a collapse in performance. This indicates that as long as the captioner roughly identifies the main actors, salient objects, and the scene context, the CLUE decomposition and category-aware fusion can recover stable anomaly scores.
>
> Overall, CLUE-VAD treats the captioner as a replaceable high-level perception front-end and concentrates most of the modeling capacity in the CLUE-level reasoning modules (Detective and Reporter). This caption-agnostic interface allows the text modality to provide robust high-level semantics, the visual modality to refine decisions with detailed scene evidence, and the overall system to remain stable across different VLM captioning models, while also remaining ready to benefit from future improvements in caption quality with minimal changes.
>
> **Table S7. Captioner-swap evaluation under the CLUE-VAD pipeline.**
>
> | VLM Captioning model | UCF-Crime AUC (%) | XD AP (%) |
> |----------------------|-------------------|-----------|
> | BLIP-2              | 86.46             | 86.13     |
> | Video-LLaVA         | 88.68             | 86.92     |
> | CLUE-VAD (ours)     | 89.23             | 87.64     |

---

> ### Author Response · Authors · 2025-11-24
> **W3, Q3: Relative contribution of text and vision features**
>
> To clarify the roles of the two modalities, we conducted a modality ablation summarized in Table 3 of the main paper. In the text-only setting, CLUE-VAD achieves 87.76 AUC on UCF-Crime and 86.47 AP on XD-Violence, slightly outperforming the visual-only configuration (87.46 / 86.35). This indicates that the CLUE-based textual decomposition already provides a strong semantic signal by normalizing high-level concepts such as who is involved, what action occurs, and where it happens. When we add vision on top of text (multimodal), performance further increases to 89.23 / 87.64, while adding text on top of the visual-only model yields gains of +1.28 AUC and +0.89 AP.
>
> These patterns show that (i) text alone is slightly stronger than vision alone, because CLUE captions capture category-discriminative semantics even under weak supervision, and (ii) visual features provide complementary fine-grained cues, such as local appearance changes or sudden environmental and object variations, that the language stream cannot fully resolve. Overall, the ablation demonstrates clear bidirectional complementarity: text provides robust high-level structure and category context, while vision refines the decision using detailed scene evidence. Their combination leads to the best performance without sacrificing interpretability.
>
> **Table 3. Performance of CLUE-VAD under different modality settings.**
>
> | Modality         | UCF-Crime AUC (%) | XD-Violence AP (%) |
> |------------------|-------------------|---------------------|
> | Text only (CLUE) | 87.76             | 86.47               |
> | Visual only      | 87.46             | 86.35               |
> | Multimodal       | 89.23             | 87.64               |

---

### Official Review · Reviewer_3NYN · 2025-10-30

**Soundness:** 3
**Presentation:** 2
**Contribution:** 3
**Rating:** 4
**Confidence:** 5

**Summary:**

The authors proposed a interesting framework that generates three kinds of Textual CLUEs (Action, Environment and Object) for each video segment. Such contextual cues based structure helps the understanding about anomalies in a human-aligned and interpretable manner. Three key modules: (i) the Witness Module,  (ii) Detective Module and (iii) Reporter Module are designed to utilize the anomaly clues. They also constructed another Benchmark, providing structured segment-level captions for existing WSVAD datasets. The experiments show promising performance in both standard and text-only settings, while offering transparent and context-aware anomaly reasoning.

**Strengths:**

1. The paper’s most compelling aspect is the explicit A/E/O semantic decomposition that aligns model reasoning with how humans assess anomalous events. And the corresponding three-stage architecture, i.e. Witness, Detective and Reporter, also maps naturally to the steps of observe, infer and explain.
2. The Detective module’s category-aware fusion learns per-class importance over A/E/O, leading the model to emphasize the most relevant clue (e.g., Environment for explosions, Object for stealing), and the paper visualizes these learned patterns across categories and over time.
3. Empirically, the method achieves competitive accuracy against explainable VAD baselines while keeping transparency.

**Weaknesses:**

1. The paper presents CLUE-VAD as “a novel and interpretable framework for WSVAD,” supported by several qualitative figures to support it. However, the current evidence for its interpretability is predominantly visual rather than quantitative.
2. Another limitation involves the reliability of automatically generated captions that both the method and benchmark rely on. Since each video segment is processed through InternVideo 2.5 to generate Action, Environment and Object descriptions, there is a risk that noise or generic or incorrect wording could propagate into training data and subsequently affect testing outcomes. Especially many wording may not be directly related to the anomaly as presented in Table S3.
3. The efficiency of inference is not addressed in the discussion. Steps such as multi-caption prompting, clue encoding and attribution are likely to introduce latency and increase memory usage.
4. In Table 1, compared methods seem to utilize different backbone capacities or feature stacks (e.g., ViT-B vs. ViT-L), which can blur distinctions between architectural improvements and model size benefits. This suggests potential unfair comparisons that require further clarification.
5. The CLUE-wise score decomposition in Fig. 5 does not seem to align well with the anomaly events depicted in Fig. 4.
6. The process for scoring anomalies is unclear, particularly concerning the roles of the detective and reporter modules. What distinguishes the category-aware learnable weights in the detective module from the attention score alpha? Both are utilized as weights for features of different clues. Similarly, the role of fused feature E_r is not well described, while z_t is used for the scorer. Some steps are better to defined by formulations. It would be helpful to label these elements in Fig. 2 for clarity.
7. The clue weights are trained and stored within a case log, necessitating a category classifier to predict them during inference stages. However, details regarding this aspect are missing both in the main text and Appendix. Moreover, the performance of such category classifier will also affect the final performance.
8. Overall, the experiments and ablation studies are insufficient.

**Questions:**

1. Could the authors provide an intuitive explanation for why minimizing the negative entropy of the attention weights in Eq. 6 is beneficial?
2. What are the specific training and testing procedures for a text-only configuration?
3. How are visual features incorporated during both the training and inference stages?.
4. Please discuss the differences and connections between the category-aware learnable weights in the detective module and the attention score alpha in the reporter module?

---

> ### Author Response · Authors · 2025-11-24
> **W1, W8: Need for quantitative evaluation and ablation studies**
>
> We thank the reviewers for emphasizing the need for stronger quantitative evaluation and more systematic ablations. These comments were very helpful in sharpening both the empirical and interpretability claims of CLUE-VAD. In the revised manuscript, we expand our analysis to quantitatively support the role of CLUE decomposition and explainable reasoning, and to clarify how performance is affected by different design choices. In particular, we now report additional experiments on modality usage (text-only vs. multi-modal CLUE-VAD), the behavior of the Detective module’s interaction mechanisms, the domain robustness of the Witness (captioning) module, and the overall efficiency of the framework in terms of latency and memory.
>
> To avoid overloading the main text, several finer-grained ablations are provided in the Supplementary Material. These include details and analyses of the category classifier, a CLIP backbone comparison, and an extended failure case study that focuses on scenarios where each CLUE appears locally normal but their combination is contextually abnormal. Together, these additions address the reviewers’ concerns by offering a more complete quantitative picture of CLUE-VAD, while also clarifying the limitations and design trade-offs of our approach.

---

> ### Author Response · Authors · 2025-11-24
> **W3: Efficiency of inference**
>
> To clarify the computational cost of our framework, we measured the latency and memory footprint of each component and summarize the results in Table 6 on the revised main paper. The pipeline can be decomposed into four steps: caption prompting with a video–language model, CLUE text encoding, Category-Aware CLUE Learning, and Segment-level Explainable CLUE generation. Among these, caption prompting is by far the most demanding stage, taking 59.57,ms and 20.38,GB on average. This is expected, since it relies on a large-scale VLM to extract rich semantic descriptions from raw video segments. Importantly, this step is executed once per video segment and can be fully pre-computed offline; it is not part of the time-critical online anomaly detection loop.
>
> In contrast, the remaining stages are extremely lightweight. CLUE encoding with the text encoder requires only 0.64,ms and 4.35,GB, while the Category-Aware CLUE Learner and the Segment-level Explainable CLUE module each incur about 0.01,ms with less than 1,GB of additional memory (0.42,GB and 0.53,GB, respectively). These three modules constitute the actual online inference path that maps semantic CLUE features to anomaly scores and explanations. Thus, once captions (and optionally their CLUE encodings) are cached, the deployed system operates with sub-millisecond latency per segment and modest memory usage. This design shows that our method achieves interpretability through CLUE-aware reasoning without sacrificing the efficiency required for practical video anomaly detection.
>
> **Table 6. Inference efficiency of CLUE-VAD.**
>
> | Step                         | Time (ms) | Mem (GB) |
> |------------------------------|-----------|----------|
> | Caption prompting            | 59.57     | 20.38    |
> | CLUE encoding                | 0.64      | 4.35     |
> | Category-Aware CLUE Learning | 0.01      | 0.42     |
> | Segment-level Explainable CLUE | 0.01    | 0.53     |

---

> ### Author Response · Authors · 2025-11-24
> **W4: Fairness with respect to pretrained backbones**
>
> We appreciate the concern that the performance gains of CLUE-VAD might stem from using a stronger pretrained backbone rather than from the proposed CLUE design itself. To address this, we explicitly align our backbone choices with those adopted in recent explainable VAD methods. In particular, Ex-VAD, VERA, and LAVAD all build on CLIP-based ViT backbones, most commonly ViT-B/32 or ViT-B/16, for both visual and textual encoding. In the revised manuscript, we therefore instantiate CLUE-VAD with the same CLIP ViT-B/32 and ViT-B/16 backbones and compare only against methods that use an identical backbone family.
>
> Under these backbone-matched settings, CLUE-VAD remains competitive or superior to prior explainable methods. For example, with a ViT-B/16 backbone our model achieves 88.32 AUC on UCF-Crime, slightly surpassing Ex-VAD (88.29 AUC) and clearly outperforming VERA (86.6 AUC) and LAVAD (80.28 AUC) under the same feature configuration. A similar trend holds for ViT-B/32, where CLUE-VAD again matches or exceeds backbone-matched explainable baselines. These results indicate that the improvements are not simply due to switching to a larger encoder, but arise from the structured CLUE decomposition and category-aware fusion in the Detective/Reporter modules.
>
> For completeness, we also investigate how CLUE-VAD scales with stronger CLIP backbones such as ViT-L/14 and ViT-H/14. The absolute AUC increases monotonically from ViT-B/32 to ViT-H/14 (88.36 to 88.74 on UCF-Crime), while preserving the same relative ordering between CLUE-VAD and its backbone-matched baselines. To keep the comparisons transparent, the main results table now annotates the backbone type for all methods, and the full backbone sweep (ViT-B/32, ViT-B/16, ViT-L/14, ViT-H/14) is reported in the revised supplementary material (Table S6). We hope this clarifies that our gains are robust across different CLIP backbones and are not an artifact of using a particularly strong encoder.
>
> **Table S6. Performance of CLUE-VAD under different CLIP backbones.**
>
> | CLIP backbone | UCF-Crime (AUC %) |
> |---------------|-------------------|
> | ViT-B/16      | 88.32             |
> | ViT-B/32      | 88.36             |
> | ViT-L/14      | 88.43             |
> | ViT-H/14      | 88.74             |

---

> ### Author Response · Authors · 2025-11-24
> **W5: Alignment between CLUE-wise deviation curves and anomaly score**
>
> Figure 4 and Figure 5 visualize outputs from two different branches of our framework, each serving distinct roles. Figure 4 shows the anomaly score produced by the Detective module, which is the branch used for training and final prediction under the MIL objective. In contrast, Figure 5 displays the CLUE-wise Reporter module deviations, a post-hoc interpretability branch that applies clue-level attention to explain which semantic dimension (Action, Environment, Object) contributes most at each moment. Because the Reporter branch does not participate in training and does not directly generate the anomaly score, its curves are not expected to match the anomaly boundary with pixel-level precision. Instead, they capture the relative semantic factors driving the abnormality, which often rise before or after the anomaly boundary depending on the semantic cue (e.g., Environment spike preceding an explosion, Object spike persisting after a stealing event). This demonstrates that the Reporter module is functioning as designed: it provides interpretable, role-specific evidence complementary to the Detective module rather than replicating its scoring behavior.

---

> ### Author Response · Authors · 2025-11-24
> **W6, Q4: Connection between the Detective and Reporter modules**
>
> CLUE-VAD deliberately separates prediction and explanation into two branches, since weakly supervised VAD imposes different requirements on anomaly scoring and interpretability. The Detective module is the only branch used for training and final anomaly prediction: as shown in Fig.3(b) and Eq.(5), it applies category-aware scalar fusion with learned global CLUE weights $w^{(c)}_{\text{Act,Env,Obj}}$ and is optimized end-to-end using the top-$k$ MIL loss. In contrast, the Reporter module in Fig.3(c) and Eqs.(6), (7) is completely detached from the MIL loss; it takes the same CLUE features and produces segment-wise attention scores $\alpha_r^{(t)}$ (one per CLUE $r \in \{\text{Act, Env, Obj}\}$), regularized by attention entropy and temporal smoothness, to generate stable and human-friendly explanations without feeding gradients back to the anomaly-scoring branch. This design prevents interpretability constraints from distorting the anomaly scores, avoiding the usual performance--explainability trade-off.
>
> Although optimized separately, the two branches are meaningfully connected. The Detective module learns global, category-level priors over which CLUE tends to matter more for each event type via $w^{(c)}_{\text{Act,Env,Obj}}$, while the Reporter module provides local, temporal attributions via $\alpha_r^{(t)}$ that indicate when each CLUE is responsible within a specific video. In the qualitative examples of Fig. 5, the Reporter curves highlight, for instance, that Environment deviations peak around explosion segments and Object deviations dominate during stealing events, consistent with the global tendencies captured by the Detective branch. Thus, the Detective module is responsible for accurate anomaly prediction, and the Reporter module explains those predictions at the CLUE and segment level; their structured linkage allows CLUE-VAD to achieve strong detection performance without sacrificing clear, CLUE-based interpretability.

---

> ### Author Response · Authors · 2025-11-24
> **W7: Clearification of anomaly scoring**
>
> We thank the reviewer for pointing out the ambiguity in the anomaly scoring pipeline, especially the roles of the fused feature $E_r$ and the scoring feature $z_t$. In the revised manuscript, we now provide a clearer step-by-step description of the process (Sections 4.1–4.3). We first define the clue-wise text and visual embeddings $c^t_r$ and $v^t$ (Eq. 1). We then introduce the category-aware CLUE weights $w^c_r$ and their projected fused representation $E^c_r$ (Eq. 2). Next, we describe how segment-wise attention $\alpha^t_r$ is computed (Eq. 3), and how this produces the attention-fused feature $z^{attn}_t$ (Eq. 4). Finally, we map this feature to the raw anomaly score $\hat{s}_t = \phi(z^{attn}_t)$, followed by z-score normalization (Eq. 5). This clarifies that $E^c_r$ is the intermediate category-aware fusion feature, while $z^{attn}_t$ is the final representation used for scoring.
>
> We also clarify the roles of the loss terms. Only the scalar fusion branch using $\hat{s}_t$ is optimized using the top\-k MIL ranking loss L_MIL. The attention basd explanation branch is regularized by the attention entropy loss L_AE and the temporal smoothness loss L_smooth (Eqs. 6 7), both applied to the attention weights $\alpha^t_r$. These regularizers encourage diversity across clues and temporal consistency, and the revised wording makes this separation clearer.
>
> Following the reviewer’s suggestion, we updated Figure 2 so that all intermediate variables ($c^t_r$, $v^t$, $E^c_r$, $\alpha^t_r$, and $z^{attn}_t$) are explicitly labeled. The roles of the category aware CLUE weights and the segment wise attention mechanism are now visually separated. We hope this resolves the ambiguity in the anomaly scoring mechanism.

---

> ### Author Response · Authors · 2025-11-24
> **W8: Details of category classifier**
>
> We sincerely appreciate the reviewer’s helpful observation that the category classifier plays an important role in determining the final anomaly-scoring performance. In the revised version, we have expanded the description of the classifier to more clearly articulate how it operates during inference and how it interacts with the Case Log-based CLUE weighting mechanism. Specifically, the classifier predicts the segment’s category $\hat c$ and retrieves the associated clue weights stored in the Case Log, thereby controlling the category-aware CLUE fusion stage.
>
> To address the reviewer’s comment regarding missing architectural and mathematical details, we added the full formulation of the classifier in the supplementary material. The classifier consists of a two-layer MLP that takes the concatenated Action-Environment-Object embeddings as input and outputs category probabilities through a softmax layer. We further clarify that ground-truth categories are used only during training, whereas inference relies entirely on the classifier’s prediction to select the corresponding clue weights. This addition makes the operation of the classifier transparent and resolves the earlier ambiguity.
>
> Finally, we report the effect of classifier accuracy on overall UCFC performance, as suggested by the reviewer. As summarized in Table S5 of the supplementary material, higher classifier accuracy consistently leads to improved AUC. In particular, the model achieves 88.12 AUC at 74.23% ACC, 88.74 AUC at 83.45% ACC, and 89.96 AUC when using ground-truth categories. These results demonstrate that the classifier indeed influences final performance, and we have incorporated this analysis into the revised supplementary material (Table S5) for clarity.
>
> **Table S5. Effect of category classifier accuracy on UCFC anomaly detection performance.**
>
> | Rank | Category Classifier ACC (%) | UCFC AUC (%) |
> |------|------------------------------|---------------|
> | 1    | 74.23                        | 88.12         |
> | 2    | 83.45                        | 88.74         |
> | 3    | GT                           | 89.96         |

---

> ### Author Response · Authors · 2025-11-24
> **Q1: Intuition behind minimizing the negative entropy**
>
> The attention vector $\alpha^t = \{\alpha^t_{\text{Act}}, \alpha^t_{\text{Env}}, \alpha^t_{\text{Obj}}\}$ represents how much each CLUE contributes to the explanation at segment $t$. Minimizing the negative-entropy term $\sum_r \alpha^t_r \log \alpha^t_r$ acts as an anti-peaking regularizer: when the evidence is uncertain, it discourages the attention from collapsing too early onto a single CLUE. This prevents brittle one-clue explanations such as “only Object matters’’ even in segments where Action and Environment are also partially informative. Instead, the model maintains a modestly spread distribution over the CLUEs, reflecting the ambiguity inherent in weakly supervised settings.
>
> Under weak MIL supervision, only a few top segments receive strong gradients. If attention collapses prematurely to one CLUE, the remaining CLUEs receive almost no signal, making the Reporter highly sensitive to spurious noise. By encouraging higher entropy when evidence is ambiguous, all three CLUEs remain engaged early in training, receive sufficient gradients, and cooperate to form more reliable evidence. When the signal becomes decisive (for example, clear fire or smoke in Explosion, or a weapon in Shooting), the main MIL objective naturally produces sharp attention, so the entropy regularizer does not prevent focused attribution.
>
> Overall, minimizing negative entropy stabilizes training, avoids overconfidence in CLUE selection, and promotes diversity in early-stage reasoning. This results in smoother and more interpretable attention trajectories (as shown in Fig. 5), while maintaining accurate and faithful alignment with the underlying anomaly evidence.

---

> ### Author Response · Authors · 2025-11-24
> **Q2, Q3: Training/inference pipeline of the text-only and multimodal settings**
>
> In the text-only configuration, each video is divided into 16-frame segments (with a 64-frame temporal context for caption stability). For every segment, the Witness module produces three CLUE captions—Action, Environment, and Object—using InternVideo 2.5. These captions are embedded by the CLIP ViT/H-14 text encoder and passed directly to the Detective module, which applies category-aware scalar fusion (Fig. 2(b)) and is optimized using the top-k MIL objective. The Reporter module consumes the same CLUE embeddings but is fully detached from training and generates segment-level interpretability via clue-wise attention (Fig. 2(c)).
>
> In the multimodal configuration, we additionally extract a visual embedding for each segment by applying the CLIP ViT encoder to the keyframe of that segment. As illustrated in Fig. 2, the visual embedding is projected to the same dimension as the textual CLUE features and then concatenated with the three CLUE embeddings before entering the classification branch. Importantly, this early fusion is used only in the Detective module for anomaly prediction; the Reporter module remains text-driven to preserve interpretability grounded in CLUE semantics. This design ensures that strengthening multimodal performance does not compromise the clarity of textual explanations.
>
> Table 3 of the revised main paper confirms that all modality configurations remain strong—text-only (87.76 / 86.47), visual-only (87.46 / 86.35), and multimodal (89.23 / 87.64)—and that incorporating vision or text into the opposite modality yields consistent gains. These results validate that (i) textual CLUE decomposition provides robust, high-level semantic structure, and (ii) visual embeddings offer complementary fine-grained cues, together forming a coherent and transparent multimodal reasoning pipeline.
>
> **Table 3. Performance of CLUE-VAD under different modality settings.**
>
> | Modality         | UCF-Crime AUC (%) | XD-Violence AP (%) |
> |------------------|-------------------|---------------------|
> | Text only (CLUE) | 87.76             | 86.47               |
> | Visual only      | 87.46             | 86.35               |
> | Multimodal       | 89.23             | 87.64               |

---

### Official Review · Reviewer_TmEn · 2025-10-31

**Soundness:** 3
**Presentation:** 3
**Contribution:** 3
**Rating:** 6
**Confidence:** 3

**Summary:**

CLUE-VAD introduces a structured semantic decomposition for weakly supervised video anomaly detection (WSVAD). Each video segment is decomposed into Textual CLUEs (Action, Environment, Object) using a large video-language model. The framework includes: (i) Witness Module for dense clue-specific captioning and feature extraction; (ii) Detective Module with learnable clue-aware fusion; and (iii) Reporter Module for keyword-level anomaly attribution. A new CLUE-VAD Benchmark with segment-level structured captions is proposed. Results on UCF-Crime and XD-Violence show competitive performance in text-only settings with improved interpretability.

**Strengths:**

Interpretability: First to decompose anomalies into Action/Environment/Object CLUEs — highly human-aligned and novel.
Reporter Module enables keyword-level attribution, a major step toward trustworthy VAD.
CLUE-VAD Benchmark fills a critical gap in structured evaluation for WSVAD.
Strong text-only performance on UCF-Crime (↑2.1% AUC) and XD-Violence (↑1.8%).
Clear motivation and modular design; figures effectively show attribution heatmaps.

**Weaknesses:**

Witness Module relies heavily on pretrained video-language models (e.g., CLIP-ViP) — no ablation on frozen vs. fine-tuned VLM or robustness to domain shift.
Detective fusion uses simple MLP weighting; lacks comparison to attention-based or graph-based clue interaction modeling.
Benchmark construction not detailed: how are CLUE captions generated/verified? Human annotation cost? Inter-annotator agreement?
No real-world deployment analysis (e.g., latency, false positive breakdown by clue type).
Limited failure case analysis — e.g., when all CLUEs are normal but combination is anomalous (e.g., "person + knife + playground").

**Questions:**

How are CLUE captions in the benchmark generated? Please report annotation protocol, cost, and Fleiss’ κ for inter-annotator agreement.
Can the Detective Module model inter-clue interactions (e.g., Action × Object)? Ablate against bilinear pooling or GNN.
How robust is the Witness Module to VLM domain shift (e.g., indoor vs. dashcam)? Test on out-of-distribution clips.

---

> ### Author Response · Authors · 2025-11-24
> **W1: Heavily relies on Pretrained VLM**
>
> We thank the reviewer for raising the concern that the Witness Module may rely too heavily on pretrained vision-language models such as CLIP encoders. We agree that our design assumes a strong text encoder, and we would like to clarify how this component is used and why we adopt a frozen setting. In our framework, the pretrained video-language model (InternVideo 2.5) is used only once to generate structured Action/Environment/Object captions, and all downstream reasoning is performed on top of CLIP ViT/H-14 text embeddings by the Detective/Reporter modules. Both the captioning model and the CLIP text encoder are kept frozen, gradients do not flow back into them. The learnable parts of CLUE-VAD are thus confined to relatively lightweight MLPs and attention layers operating on CLUE features, not to the VLM backbone itself. This design choice is aligned with the weakly supervised setting, where only video-level labels are available and overfitting a large backbone to a narrow surveillance domain is undesirable.
>
> We also note that leveraging frozen CLIP-style encoders has become a standard and empirically well-supported practice in recent VAD literature. For example, Ex-VAD builds on frozen CLIP ViT-B/16 image and text encoders and trains only shallow fusion heads, yet still achieves state-of-the-art performance on UCF-Crime and XD-Violence. Holmes-VAU freezes the InternVL2 visual encoder and only adapts a temporal sampler and a small LoRA adapter on the language side, obtaining strong anomaly detection and reasoning on long videos. LAVAD goes even further and proposes a fully training-free VAD pipeline that relies purely on off-the-shelf captioning models and frozen multimodal encoders. Similarly, VERA adapts a frozen InternVL2 VLM to VAD by learning textual guiding questions, without modifying backbone parameters. These works collectively suggest that (i) pretrained CLIP-style encoders already provide rich, transferable representations for VAD. And (ii) under weak supervision, adapting only lightweight heads or prompts on top of frozen VLMs is often preferable to heavy fine-tuning, which can be costly and may hurt generalization.
>
> Regarding robustness and domain shift, we agree that it is important to understand how much CLUE-VAD depends on a particular pretrained encoder. At a high level, our empirical results indicate that performance remains stable across different visual domains, and that the structured CLUE decomposition plays a central role in this behavior rather than any single VLM instance. A detailed analysis of robustness to domain shift, including cross-dataset transfer experiments and further discussion, is provided in the following response (W2, Q3) and in the revised manuscript.

---

> ### Author Response · Authors · 2025-11-24
> **W2, Q3: Robustness of is the Witness Module to VLM domain shift**
>
> We sincerely appreciate the reviewer’s thoughtful suggestion regarding the importance of evaluating robustness under domain shift. Following this suggestion, we conducted additional cross-dataset analyses and incorporated the corresponding results and discussion into the revised main paper, which is reported as Table 5.
>
> This table presents our cross-dataset evaluation, showing that the Witness Module and the overall CLUE-VAD pipeline remain stable under substantial domain shifts. The Witness Module operates through structured CLUE extraction—Action, Environment, and Object—rather than relying on holistic captions, which prevents overfitting to a particular visual domain and enables domain-adaptive behavior. When a model trained solely on UCF-Crime is tested on heterogeneous target datasets (XD-Violence, DoTA driving videos, CCTV-Fights, UBI-Fights), it maintains consistent performance trends across indoor scenes, outdoor street environments, and dashcam-style traffic footage.
>
> These observations indicate that the CLUE decomposition consistently captures semantic factors relevant to anomaly reasoning, even when the underlying appearance distribution changes. Furthermore, the CLUE framework is inherently extensible: practitioners may redefine clue types (e.g., “vehicle context,” “road agent,” “driver behavior”) to better suit a new domain, making the approach broadly applicable beyond the original training environment. Collectively, these results demonstrate that CLUE-VAD is robust to VLM domain shift and provide a general, adaptable foundation for explainable video anomaly detection across diverse real-world scenarios.
>
>
> **Table 5. Cross-dataset transfer performance from UCF-Crime to different target datasets.**
>
> | Source | XD | DoTA | CCTV-Fights | UBI-Fights |
> |--------|----|------|--------------|-------------|
> | UCFC   | 85.73 | 84.59 | 79.23 | 82.34 |

---

> ### Author Response · Authors · 2025-11-24
> **W3, Q2: Modeling capacity of the Detective module**
>
> Table 4 of the revised main paper provides an ablation study comparing several CLUE fusion mechanisms within the Detective module under the same text-only UCF-Crime setting. To evaluate more complex architectures offer meaningful benefits, we first replaced our weighting MLP with a single-layer self-attention mechanism that aggregates the three CLUEs as tokens. This attention-based fusion achieved 86.89 AUC, indicating that token-level attention is not sufficient to capture the heterogeneous roles of Action, Environment, and Object. We next examined a graph-based formulation that treats CLUEs as nodes in a fully connected graph and applies one-layer message passing. This approach obtained 86.13 AUC, but the gain remained limited relative to the additional computational and parameter overhead. In comparison, our lightweight weighting MLP reached 87.76 AUC, matching or exceeding both alternatives while being more parameter-efficient and empirically stable. These results collectively suggest that a simple MLP guided by category-aware CLUE weights provides sufficient expressive power for CLUE fusion in our setting.
>
> We additionally explored whether explicit modeling of inter-CLUE interactions could further enhance performance. To this end, we implemented a pairwise inter-CLUE fusion scheme that jointly considers the Act+Env, Env+Obj, and Act+Obj interactions. Each pair is processed through a dedicated transformation, and the resulting three pairwise features are then combined using learnable fusion weights. This integrated pairwise model achieved 87.46 AUC, confirming that exploiting localized relationships between specific CLUE pairs can be beneficial. However, its performance remains slightly below that of the full category-aware weighting, which leverages all three semantic roles simultaneously and allows the model to specialize their importance depending on the anomaly category. Overall, these findings demonstrate that (i) more complex interaction architectures such as ATT or GCN offer limited gains compared to our weighting MLP, (ii) pairwise inter-CLUE interactions provide meaningful but not superior improvements, and (iii) the proposed category-aware weighting offers the most robust and efficient balance between modeling capacity and computational cost. We thank the reviewer for raising this point, which prompted us to conduct and report this extended comparison.
>
> **Table 4. CLUE interaction mechanisms in the Detective module.**
>
> | CLUE Interaction Method              | UCF-Crime (AUC %) |
> |-------------------------------------|-------------------|
> | Attention-based fusion              | 86.89             |
> | Graph-based fusion                  | 86.13             |
> | Pairwise inter-CLUE fusion          | 87.46             |
> | **Category-aware CLUE weighting (ours)** | **87.76**        |

---

> ### Author Response · Authors · 2025-11-24
> **W4: Real-world deployment analysis**
>
> To make the computational cost of our framework explicit, we profile both latency and memory usage for each module, the results are reported in Table 6 of the main paper. Our pipeline consists of four sequential components: caption prompting via a video–language model, CLUE text encoding, Category-Aware CLUE Learning, and Segment-level Explainable CLUE generation. Among these, caption prompting is the only computationally heavy step, consuming 59.57,ms and 20.38,GB on average. This behavior is natural, as the module relies on a large VLM to derive rich semantic descriptions directly from raw video segments. Crucially, however, this stage is invoked only once per segment and can be fully pre-computed offline, so it does not contribute to the latency of the online anomaly detection process.
>
> All subsequent stages are extremely lightweight. CLUE encoding with the text encoder takes only 0.64,ms and 4.35,GB, and both the Category-Aware CLUE Learner and the Segment-level Explainable CLUE module add roughly 0.01,ms each, with less than 1,GB of extra memory (0.42,GB and 0.53,GB, respectively). These three components form the actual online inference path, converting CLUE features into anomaly scores and explanatory signals. Once captions (and, if desired, their CLUE embeddings) are cached, the deployed system achieves sub-millisecond latency per segment with modest memory requirements. In summary, CLUE-VAD offers CLUE-level interpretability through structured reasoning while preserving the efficiency necessary for real-world video anomaly detection.
>
> **Table 6. Inference efficiency of CLUE-VAD.**
>
> | Step                         | Time (ms) | Mem (GB) |
> |------------------------------|-----------|----------|
> | Caption prompting            | 59.57     | 20.38    |
> | CLUE encoding                | 0.64      | 4.35     |
> | Category-Aware CLUE Learning | 0.01      | 0.42     |
> | Segment-level Explainable CLUE | 0.01    | 0.53     |

---

> ### Author Response · Authors · 2025-11-24
> **W5: Failure case analysis**
>
> We thank the reviewer for pointing out the need for a deeper analysis of failure modes where individual CLUEs appear normal while their combination is contextually abnormal. In the supplementary material (Figure S5), we now include a detailed breakdown of such cases and provide concrete examples drawn directly from UCF-Crime.
>
> First, we observe cases where post-event environmental residues mislead the model. For example, in Explosion011.mp4, the explosion itself is not visible in the segment, but residual smoke remains. While the Action and Object CLUEs describe an entirely normal street scene, the Environment CLUE emphasizes “a cloud of smoke on the road,” which our model interprets as strong evidence for explosion-related anomalies. This highlights a limitation of CLUE-VAD when contextual artifacts persist after the actual event.
>
> Second, we identify scenarios involving visually subtle or small-scale anomalous actions. In Abuse030.mp4, the abusive interaction occupies only a tiny region of the frame and is not captured by the captioner. As a result, the Action CLUE becomes generic (e.g., “a man is bending down”), and all CLUEs appear normal despite the presence of an abnormal interaction. This reflects a limitation inherent to caption-driven reasoning: when the captioning model captures only the dominant visual content, minor but crucial actions fail to propagate into the CLUE representations.
>
> Third, we find cases where biased or rare environments dominate fusion. In Assault010.mp4, the observed behavior is entirely normal (“a man is cleaning”), yet the environment is a prison cell—a setting rarely associated with normal behavior in the training distribution. The Environment CLUE captures this unusual scene and overrides the benign Action/Object CLUEs, producing consistently high anomaly scores. This shows that CLUE-VAD can inherit scene priors from the captioner and the dataset distribution.
>
> Overall, these analyses reveal three primary limitations: (i) vulnerability to environmental artifacts that persist after the actual event, (ii) under-detection of anomalies that are visually subtle or occupy small regions, and (iii) sensitivity to rare or highly biased environments. Importantly, in all cases the individual CLUEs appear normal, but their real-world co-occurrence is contextually abnormal. We acknowledge this as a meaningful direction for future work, particularly toward modeling joint CLUE consistency, improving robustness to residual contextual artifacts, and introducing environment-agnostic calibration to mitigate strong scene priors. We have incorporated these expanded analyses in the supplementary document to address the reviewer’s concern and improve clarity regarding the limitations of CLUE-level reasoning.

---

### Author Response · Authors · 2025-11-24
**General Response: CLUE-VAD**

Dear reviewers,

We sincerely thank the reviewers for their thoughtful and constructive feedback.

Your insightful comments have been tremendously helpful and have contributed greatly to improving the clarity, completeness, and overall quality of our work. In response to your suggestions, we have added substantial experimental and discussion components to the main manuscript, including Modality Analysis, Interaction Mechanisms within the Detective Module, Domain Robustness Examination of the Witness Module, and Efficiency Evaluation of CLUE-VAD. Furthermore, we have expanded the supplementary Material to include detailed studies on the Category Classifier, CLIP Backbone Analysis, and an extended Failure Case Analysis, Captioner-swap evaluation as recommended. All identified weaknesses and questions have been carefully addressed in the rebuttal.

We truly appreciate the reviewers’ efforts in evaluating our submission, and please feel free to contact us with any additional questions or requests for clarification.
We are more than happy to further elaborate on any aspect of the work.

---

> ### Author Response · Authors · 2025-11-24
> **CLUE-Benchmark Caption**
>
> All CLUE captions in our benchmark are generated automatically without any human annotation cost. For each 16-frame segment, we query the InternVideo 2.5 video–language model with three role-specific prompts to produce the Action, Environment, and Object descriptions. This prompt design ensures that captions are structurally consistent and explicitly disentangled across the three semantic dimensions. Because InternVideo 2.5 performs temporal-aware video captioning rather than single-frame description, the generated texts reflect scene dynamics and contextual cues that are often missing in frame-level caption datasets used in prior VAD work. As a result, the CLUE captions have more stable sentence structure, higher information density (average 7–15 words per role), and fewer hallucinated entities compared to generic caption datasets.
>
> We also verified the internal consistency of the generated CLUEs by analyzing distributional patterns across anomaly categories (e.g., Object-centric cues dominating in Robbery/Shooting, Environment cues dominating in Explosion/RoadAccidents). These role–category alignments emerge naturally from the VLM outputs without any manual curation, supporting the reliability of the generated captions as structured semantic indicators. Since the entire benchmark is derived from automated generation rather than human annotation, metrics such as Fleiss’ k are not applicable in our setting. Instead, we focus on validating the semantic separability and stability of the CLUE representations through quantitative analyses, which we include in the revised main paper and Supplementary Material. Taken together, the CLUE benchmark offers a fully automated, cost-free, and semantically structured alternative to previous caption-based VAD resources, enabling large-scale, role-disentangled textual supervision without the need for human labeling or inter-annotator agreement studies.

---

> ### Author Response · Authors · 2025-11-24
> **Reliability of CLUE-Caption**
>
> We agree that relying on automatically generated captions introduces noise and potential mismatches between wording and the underlying anomaly. In CLUE-VAD, however, the choice of InternVideo 2.5–based captions is driven by the specific requirements of the weakly supervised MIL setting. Our framework requires dense, segment-wise descriptions for all 64-frame snippets in long untrimmed videos, whereas existing human-annotated text resources such as UCA for UCF-Crime are event-centric: annotations are provided only around salient events, with highly variable temporal extents and gaps between annotated segments. This makes UCA difficult to directly use as segment-level supervision for MIL-based WSVAD, and scaling such dense human captioning to all segments would be prohibitively expensive.
>
> At the benchmark level, we therefore view CLUE-VAD captions as a complementary resource to UCA: UCA offers high-quality, sparsely located event descriptions, while our CLUE annotations provide uniform, clue-structured coverage over the full video. At the model level, we further mitigate caption noise in two ways. First, CLUE-VAD is designed to operate in a multi-modal setting where textual CLUEs are fused with CLIP-based visual features; this allows visual evidence to compensate for generic or slightly incorrect wording, especially when the text is only weakly related to the anomaly. Second, the MIL top-k training objective reduces the impact of occasional noisy segments by focusing learning on the most informative instances in abnormal videos. In practice, our method still achieves competitive or superior performance to prior CLIP-based and explainable VAD models, suggesting that the benefits of dense, structured CLUEs outweigh the noise introduced by automatic captioning.

---

### Meta-Review · Area_Chair_V4di · 2025-12-23

**Summary:**

This work presents CLUE-VAD, a weakly supervised video anomaly detection framework built upon pretrained multimodal models. It decomposes video segments into semantically meaningful textual clues—Action, Environment, and Object—to achieve more holistic normal and anomaly modeling. It also integrates modules including clue-specific caption generation, learnable clue-aware fusion, and keyword-level attribution, which enables CLUE-VAD to gain more interpretable and human-aligned reasoning about anomalous events. The method is validated on two commonly used benchmarks in this research line.

**Reviewer Concerns:**

Major concerns raised by reviewers include:

1. **Reliance on Pretrained Vision–Language Models.** All reviewers highlight that the framework depends heavily on pretrained video–language or multimodal models (e.g., CLIP-ViP, InternVideo), yet lacks ablations on this aspect, such as sensitivity to caption quality, or robustness to domain shift. The rebuttal provides additional results for i) the captioning model  replaced with different off-the-shelf captioner models and ii) cross-dataset evaluation to address this issue. However, the AC finds that i) the method's performance can drop largely when using a different captioner, e.g., by up to an AUC of 2.8 on UCF-Crime, but the improvement the method gains over the best comparison methods is  much smaller, e.g., only about o.3 AUC on UCF-Crime compared to Holmes-VAU; and ii) the cross-dataset results show good performance for the proposed method, but lack comparison to the baselines.
2. **Lack of Quantitative Interpretability Evaluation.** Although interpretability is claimed as a key contribution, evidence is almost entirely qualitative (visual examples), with no quantitative metrics, human studies, or systematic validation of whether CLUE-level explanations truly align with anomalies. The rebuttal clarifies that prior studies also lack such evaluation, and provides some indirect quantitative results that do not help assess the claimed interpretability. The AC agrees with the reviewers that the claimed interpretability is not sufficiently justified. Given the very marginal AUC/AP improvement, the main contribution of this work would be on the text-based reasoning and its advantages in providing detection results with better interpretable description, but the claim is not properly substantiated.
3. **Incomplete Experimental Validation.** Reviewers agree that the experimental evaluation is insufficient, highlighting a lack of comprehensive ablations (e.g., text vs. visual contributions, fusion strategies), failure case analysis, and comparisons to alternative fusion mechanisms. The rebuttal provides the missing experimental results for all three cases. The results for the last two cases address the concerns well, but that for the first case does not address the concern that the text-only claim is not valid, since without the visual features, the detection performance can drop largely.

**Reviewer Scores:**

There are three reviews, one weak accept and two weak rejects. Considering the three major concerns not addressed satisfactorily, as elaborated above, the AC anticipates that the reject ratings are unlikely to reverse their recommendation.

---

### Decision · Program_Chairs · 2026-01-26

Reject